# STRONG MODEL COLLAPSE

**Elvis Dohmatob**[1,2,3]**, Yunzhen Feng**[4,†]**, Arjun Subramonian**[5,†]**, Julia Kempe**[1,4]
[1]Meta FAIR    [2]Concordia University    [3]Mila    [4]NYU    [5]UCLA
[†]Work done while interning at Meta. Correspondence to `elvis.dohmatob@concordia.ca`

## ABSTRACT

Within the scaling laws paradigm, which underpins the training of large neural networks like ChatGPT and Llama, we consider a supervised regression setting and establish a strong form of the model collapse phenomenon, a critical performance degradation due to synthetic data in the training corpus. Our results show that even the smallest fraction of synthetic data (e.g., as little as 1 per 1000) can still lead to model collapse: larger and larger training sets do not enhance performance. We further investigate whether increasing model size, an approach aligned with current trends in training large language models, exacerbates or mitigates model collapse. In a simplified regime where neural networks are approximated via random projections of tunable size, we both theoretically and empirically show that larger models can amplify model collapse. Interestingly, our theory also indicates that, beyond the interpolation threshold (which can be extremely high for very large datasets), larger models may mitigate the collapse, although they do not entirely prevent it. Our theoretical findings are empirically verified through experiments on language models and neural networks for images.

## 1  INTRODUCTION

The term Model Collapse refers to a critical degradation in the performance of AI models, particularly when a significant portion of their training data consists of synthetic data generated by other models. As detailed in Shumailov et al. (2023), this phenomenon arises as the model gradually overfits to patterns found in synthetic data, which may not fully represent the richness or variability of real-world data. Over successive training cycles, this feedback loop results in the model reinforcing errors, biases, or oversimplifications from the synthetic data. Consequently, the model's ability to generalize to real-world data is compromised, as it increasingly relies on the distorted distribution provided by prior AI generations rather than learning accurate representations of the real world.

This phenomenon was observed empirically (Hataya et al., 2023; Martínez et al., 2023a;b; Bohacek & Farid, 2023; Briesch et al., 2023; Guo et al., 2023) and described theoretically (Alemohammad et al., 2023; Bertrand et al., 2023; Dohmatob et al., 2024a;b). The connection to the breakdown of neural scaling laws (Kaplan et al., 2020) has been pointed out and analyzed in Dohmatob et al. (2024b): as data becomes more synthetic, larger training sets do not enhance performance.

The issue is especially concerning in large-scale AI systems like ChatGPT and Llama  (Touvron et al., 2023; Dubey & et al., 2024), which rely heavily on vast amounts of training data to maintain their performance. If synthetic data is used in training these models, even in small quantities, the model can start producing "gibberish" or nonsensical outputs, contains misinformation, or reflect stereotypes. This is because the model effectively starts to amplify its own mistakes (Shumailov et al., 2024). This feedback loop results in a gradual loss of model fidelity, reducing its ability to generalize or adapt to new, unseen test environments.

### 1.1  MAIN CONTRIBUTIONS

In this work, we establish a series of results which shed more light on model collapse, bringing the phenomenon closer to a solid theoretical foundation. We consider the following important questions:

(Q1) *Is model collapse inevitable or can it be fixed by strategically mixing synthetic and real data?*

(Q2) *Are larger models more prone to model collapse than smaller ones?*

Our theoretical analysis focuses on the solvable setting of linear regression with and without random projections, with the latter serving as an approximation of neural networks by means of random feature maps (Maloney et al., 2022; Bach, 2023). Also, in accordance with the current "neural scaling laws" paradigm (Kaplan et al., 2020; Hoffmann et al., 2022) whichs underlies the training of LLMs, where models and dataset sizes become larger over time, we focus on the setup where the total amount of data (synthetic + real data) used for training grows arbitrarily.

Let us summarize our main findings.

*– Result #1: Strong Model Collapse.* First, we establish a robust negative result which shows that model collapse generally persists even on a mixture of synthetic and real training data, as long as the fraction of training data which is synthetic does not vanish (cf. Sections 3.1 and 3.2). By synthetic data, we mean any training data from a distribution which deviates from the distribution of real data, i.e. data on which the test performance is evaluated. Thus, model collapse cannot generally be mitigated by simple adjustments such as data weighting (Jain et al., 2024). On the other hand, we show (Section 5) that sophisticated iterative mixing strategies like Ferbach et al. (2024) can mitigate model collapse. However, apart from additional computational overhead, such a strategy requires access to a supply of clean/real data whose size grows at least at same rate as the synthetic data.

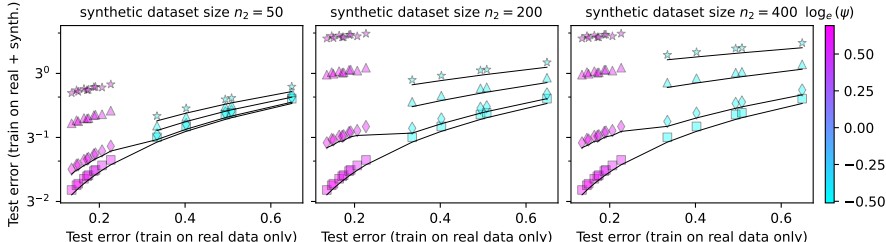

Figure 1: **Pareto diagram: Understanding the role of model size in model collapse**. We compare the test error (on the real / true data distribution), for a random projections model (Equation (5) of Section 2.2) when training is done on a mix of synthetic and real data (y-axis), versus real data only (x-axis); in both cases, the total amount of training data is fixed to $n = 500$. On the scatter plots, square points correspond to very high-quality synthetic data (i.e from a distribution which is close to the true data distribution), diamonds correspond to high-quality synthetic data, triangles correspond to low-quality, while stars correspond to very low-quality synthetic data. The black lines correspond to the Pareto frontiers for each level of quality of the synthetic data; the higher the frontier above the diagonal in the given setting, the more serious is the model collapse. The colorbar is the log of the parametrization rate $\psi = m/n$, where $m$ captures the size of the model.

*– Result #2: Model Size and Model Collapse.* In Section 3.2, we disentangle the effect of a model's size on its ability to cope with model collapse. We show that in general, bigger models will suffer more from model collapse as soon as the deviation between the distribution of the synthetic data and real data is significant. Crucially, our theory also predicts that past the interpolation threshold point, this tendency can be reversed: large models become more robust to model collapse. Put together, these results predict the existence of a double-descent curve regarding the model collapse phenomenon. This is illustrated in Figures 1 and 2. Thus, the model collapse profile depends critically on design choices like model size.

**Experimental Validation.** Our theoretical results are empirically confirmed with experiments in:

- Toy settings, including random projections model on Gaussian data, and shallow networks fully trained on the MNIST dataset (Deng, 2012). Refer to the end of Section 3.2 and Appendix A.2.
- Realistic setting of GPT-2 models trained on BabiStories (Zhang et al., 2024a), a reproduction of TinyStories (Eldan & Li, 2023) using the Mixtral-8x7B open language model (Jiang et al., 2024)). Refer to Section 4.

**Approach.** From a technical standpoint, our theoretical analysis focuses on regression problems in the classical linear setting introduced in Dohmatob et al. (2024a) for studying model collapse, and also the setting of neural networks in a simplified regime which can be approximated by random projections (Maloney et al., 2022; Bach, 2023). We employ the tools of operator-valued free probability theory (OVFPT) (Mingo & Speicher, 2017) to obtain a new bias-variance decomposition $E_{test} \simeq B + V + \zeta$, of the test error evaluated on the real / true data distribution, of a model trained on a mixture of real and synthetic data. The extra term $\zeta$ then induces model collapse.

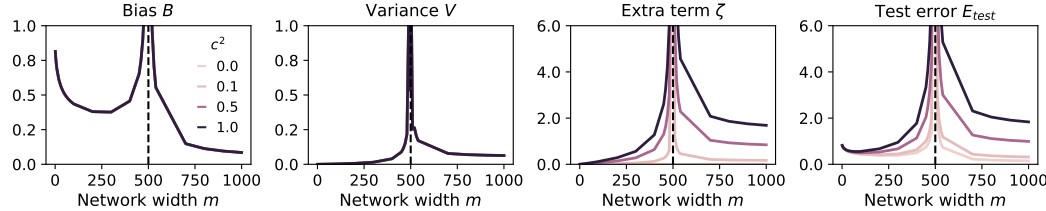

Figure 2: **Illustration of our new bias-variance decomposition** $E_{test} \simeq B + V + \zeta$ for neural networks in the simplified random projections regime (cf. Section 3.2), trained on a mixture of real and synthetic data. The sum $B + V$ corresponds to the classical bias variance decomposition in this setup when all the training data is real. The extra term $\zeta$ is responsible for model collapse when training is done on a mixture of real and synthetic data. The scalar $c^2$ characterizes the quality of the synthetic data (cf. Definition 1), via its mismatch with the real data distribution. The vertical line corresponds to the interpolation threshold $m = n$, where $m$ is the model size and $n$ is the total sample size. Notice the well-known double-descent curve in the bias.

### 1.2 RELATED WORK

The theoretical study of model collapse in the setting of high-dimensional supervised-learning with linear regression and kernel ridge regression was initiated in Dohmatob et al. (2024a). This work derives analytical formulas that quantitatively describe iterative retraining on synthetic data in both under-parameterized and over-parameterized regimes, considering both low- and high-dimensional asymptotics. It places itself within an important body of works studying kernel ridge regression (on "clean" data), which serves as an effective proxy for neural networks in various regimes, for instance in the infinite-width limit (Neal, 1996; Williams, 1996; Jacot et al., 2018; Lee et al., 2018) or in the lazy regime of training (Chizat et al., 2019) and are a testbed to study interesting phenomena observed in deep learning. For instance, (Rahimi & Recht, 2008; Rudi & Rosasco, 2017; Maloney et al., 2022) study scaling laws for regression in the random feature model and (Bach, 2023) analyses double descent in this setting. Scaling laws have been shown for kernel models under the Gaussian design, e.g. in Caponnetto & de Vito (2007); Spigler et al. (2020); Cui et al. (2022) for regression and (Cui et al., 2023) for classification.

Very few theoretical works tackle the analysis of models trained on mixtures of original (real / clean) and synthetic data. Bertrand et al. (2023) analyze the training process at the distribution level and provide stability results under a locality assumption in parameter space. Seddik et al. (2024) analyze the mixing of discrete original and synthetic data, and provide upper bounds on the amount of synthetic data that can be included to avoid model collapse. Let us also mention the recent works (Jain et al., 2024; Ferbach et al., 2024) which are potential methods for mitigating model collapse. Jain et al. (2024) analyze linear regression on *isotropic* Gaussian data for mixtures of clean and synthetic data by minizing a strategically weighted sum of losses (one term for each data source, real and synthetic), while Ferbach et al. (2024) can be seen as a multi-step version thereof where at each stage, the synthetic data generator is distilled by interpolating with real data. These methods are analyzed in Section 5, where we outline their shortcomings regarding model collapse.

Finally, a few works go beyond the mixing scenario and analyze how to curate or filter synthetic data to avoid model collapse (Feng et al., 2024; Zhang et al., 2024b; Alemohammad et al., 2024; Gillman et al., 2024; Yang et al., 2025; Askari-Hemmat et al., 2025), but a rigorous study of their effectiveness is still lacking.

## 2 THEORETICAL SETUP

### 2.1 DATA DISTRIBUTIONS

Consider an iid sample from $\mathcal{D}_1 = \{(x_i, y_i) \mid 1 \le i \le n_1\}$ of size $n_1$ from the true data distribution $P_1$ and an independent iid sample $\mathcal{D}_2 = \{(x_i, y_i) \mid n_1 + 1 \le i \le n\}$ of size $n_2$ from another data distribution $P_2$ (which we shall hereafter call the *synthetic data distribution*), where $n := n_1 + n_2$ is the total amount of training data. Here, $P_k = P_{\Sigma_k, w_k^*, \sigma_k^2}$ is the distribution on $\mathbb{R}^d \times \mathbb{R}$ given by

> **(Features)** $x \sim N(0, \Sigma_k)$,
> **(Labels)** $y = x^\top w_k^* + \epsilon$, with $\epsilon \sim N(0, \sigma_k^2)$ independent of $x$. 
> (1)

Each $\Sigma_k$ is a $d \times d$ positive-definite covariance matrix which captures the intrinsic variations of the input feature vector $x$. The $\sigma_k$'s control the level of label noise in each distribution.

**Structure of the Label Shift.** For conciseness, we will assume the following priors on the $w_k^*$'s

- *True labelling function: $w_1^* \sim N(0, \Gamma)$,*
- *Mismatch between real and synthetic: $\delta := w_2^* - w_1^* \sim N(0, \Delta)$, independent of $w_1^*$,*

for some $d \times d$ positive-semidefinite matrices $\Gamma$ and $\Delta$.

**Remark 1.** *To ease the presentation of our results, we shall assume that the matrices $\Sigma_1$, $\Sigma_2$, $\Gamma$, and $\Delta$ are diagonal matrices, and therefore commute. Furthermore, except otherwise explicitly stated, we shall assume equal covariance matrices, and take $\Sigma_1 = \Sigma_2 = \Sigma$ as in Dohmatob et al. (2024a).*

The matrix $\Gamma$ captures the structure of the ground-truth labelling function in the real / test distribution $P_1$. Together with the label-noise levels $\sigma_1^2$ and $\sigma_2^2$, the matrix $\Delta = \text{cov}(w_2^* - w_1^*)$ captures the covariance structure of the disparity between the true data distribution $P_1$ and the synthetic data distribution $P_2$ regarding the conditional distribution $p(y|x)$; the marginal distribution of $x$ stays the same under $P_1$ and $P_2$ due the assumption $\Sigma_1 = \Sigma_2 = \Sigma$. For example, the self-consuming-loops setup of Dohmatob et al. (2024a) corresponds to taking $\Delta$ proportional to the precision matrix of the input features $\Sigma^{-1}$. Thus, the size of the fluctuations of each component $\delta_j$ of the difference $w_2^* - w_1^*$ is inversely proportional to the standard deviation of the corresponding feature. Another important setup is the case where the fluctuations are isotropic, i.e taking $\Delta \propto I_d$.

**Quality of Synthetic Data.** Due to the a priori general structure of $\Delta$, the label corresponding to an input $x$ will be different for both distributions, even in the absence of label-noise. On average, the $L_2$-norm of this difference is $\mathbb{E}_{w_1^*, w_2^*} \mathbb{E}_{x \sim N(0, \Sigma)} [(x^\top w_1^* - x^\top w_2^*)^2] = \text{tr}\, \Sigma \Delta$. We therefore define

**Definition 1.** *The quality of synthetic data is defined as $c^2(\Delta) = (1/d) \text{tr}\, \Sigma \Delta$, which captures the disparity between the synthetic data distribution $P_2$ and the real data distribution $P_1$ (small values of $c^2(\Delta)$ are better). For example, if $\Delta = c^2 \Sigma^{-1}$ as in Dohmatob et al. (2024a), then $c^2(\Delta) = c^2$.*

## 2.2 MODELS AND PERFORMANCE MEASURE

Given this training data, the goal of a learner is to construct an estimator $\widehat{w}$. This can be seen as a linear model from $x \mapsto x^\top \widehat{w}$. Evaluated on the real / true data distribution $P_1$ (which coincides with the distribution from which the real component $\mathcal{D}_1$ of the training dataset $\mathcal{D}$ is drawn), the test error of a model $\widehat{f} : \mathbb{R}^d \to \mathbb{R}$ is defined by

$$E_{test}(\widehat{f}) = \mathbb{E}_{\mathcal{D}} \mathbb{E}_{x \sim N(0, \Sigma_1)} [(\widehat{f}(x) - x^\top w_1^*)^2]. \tag{2}$$

This will be our main object of study, for different models $\widehat{f}$. The outermost expectation $\mathbb{E}_{\mathcal{D}}$ is to quench the randomness in the training dataset $\mathcal{D}$ used to train the model.

We consider two families of analytically tractable models: (1) classical linear models obtained via penalized regression in the input space, and (2) models obtained via penalized regression in a feature space given by random projections. The latter allows us to study the role of model size in model collapse, by varying the output dimension of the random projection mapping. This output dimension $m$ controls the size of a neural network in a simplified regime (Maloney et al., 2022; Bach, 2023).

**(1) Classical Linear Model.** We start with a setup motivated by Dohmatob et al. (2024a). We are interested in the penalized linear model (ridge) $\widehat{f}_{CL} : x \mapsto x^\top \widehat{w}$ with parameter vector $\widehat{w}$ given by

$$\widehat{w} = \arg \min_{w \in \mathbb{R}^d} \frac{1}{n} \sum_{i=1}^{n} (x_i^\top w - y_i)^2 + \lambda \|w\|^2, \tag{3}$$

trained on the total dataset $\mathcal{D} = \mathcal{D}_1 \cup \mathcal{D}_2$. Of course, the unregularized limit $\lambda \to 0^+$ corresponds to ordinary least-squares (OLS). We shall work in the following so-called proportionate scaling limit

*(Proportionate Scaling Limit for Classical Linear Model)* For fixed $\phi \in (0, \infty), p_2 \in (0, 1)$,

$$d, n, n_1, n_2 \to \infty, \quad n_2/n \to p_2, n_1/n \to p_1 = 1 - p_2, \quad d/n \to \phi. \tag{4}$$

The extreme cases $p_1 \to 0^+$ and $p_2 \to 0^+$ correspond to training on only synthetic (resp. real) data. In particular, $p_1 \to 0^+$ corresponds to the setting considered in Dohmatob et al. (2024a). Note that in the isotropic setting where $\Sigma \propto I_d$, $\phi$ controls the speed of learning on clean data. Indeed, for

small $\phi$, the scaling law in this case is known (Hastie et al., 2022) to be $E_{test} \simeq \sigma_1^2 \phi + O(\phi^2)$. As we shall see (Corollary 1), this scaling law gets deformed in the presence of synthetic data in the training dataset, leading to model collapse.

**(2) Random Projections Model.** We consider neural networks in a simplified regime which can be approximated via random projections (Maloney et al., 2022; Bach, 2023), i.e $f(x) = x^\top S v$. Here, $S$ is a $d \times m$ random matrix with iid entries from $N(0, 1/d)$; it maps an input-vector $x \in \mathbb{R}^d$ to a random feature vector $z = \Phi(x) := S^\top x \in \mathbb{R}^m$. Only the "read-out" weights $v \in \mathbb{R}^m$ are learned, by fitting on the dataset $\mathcal{D}$. Consider the model $\widehat{f}_{RP} : x \mapsto \Phi(x)^\top \widehat{v}$, where $\widehat{v}$ is given by

$$\widehat{v} = \arg\min_{v \in \mathbb{R}^m} \frac{1}{n} \sum_{i=1}^{n} (v^\top \Phi(x_i) - y_i)^2 + \lambda \|v\|^2. \tag{5}$$

Note that such a simplified neural network model has been proposed in the literature as a theoretical testbed for studying intriguing properties of neural networks, like scaling laws (Maloney et al., 2022) and double-descent (Bach, 2023). Also see Section 1.2. It can be shown that the extreme case $m/n \to \infty$ reduces to the classical linear model.

We shall work in the following asymptotic regime:

> *(Proportionate Scaling Limit for Random Projections Model)*
>
> $$d, m, n, n_1, n_2 \to \infty, \quad n_1/n \to p_1, n_2/n \to p_2, \quad d/n \to \phi, m/d \to \gamma, m/n \to \psi, \tag{6}$$
>
> for some constants $\phi, \gamma, \psi \in (0, \infty)$ and $p_1, p_2 \in (0, 1)$, with $p_1 + p_2 = 1$ and $\psi = \phi\gamma$.

Note that the ratio $\psi/\phi \simeq md$ captures the size of the network, though the number of trainable parameters (the read-out layer) is $m \simeq \gamma d$.

# 3 A NEW BIAS-VARIANCE DECOMPOSITION AND THE EMERGENCE OF STRONG MODEL COLLAPSE

## 3.1 CLASSICAL LINEAR MODELS

We begin with an analysis of the test error $E_{test}(\widehat{f}_{CL})$ for the classical linear model defined in (3) trained on a mixture of synthetic and true / real data, but evaluated on test data from the true data distribution only. We will establish a new bias-variance decomposition with an additional term which quantitatively reveals the emergence of model collapse (Shumailov et al., 2023; 2024).

Let us first recall some standard notations. For any $t \geq 0$ and integer $k$, let $\mathrm{df}_k(t; \Sigma)$ be the order-$k$ degrees of freedom of $\Sigma$ at $t$, defined by $\mathrm{df}_k(t; \Sigma) := \mathrm{tr}\, \Sigma^k (\Sigma + t I_d)^{-k}$. Define

$$u = u(n, \lambda; \Sigma) := \frac{\mathrm{df}_2(\kappa; \Sigma)/n}{1 - \mathrm{df}_2(\kappa; \Sigma)/n}, \tag{7}$$

where $\kappa = \kappa(n, \lambda; \Sigma)$ is the unique positive solution to the fixed-point equation

$$\kappa - \lambda = \kappa \, \mathrm{df}_1(\kappa; \Sigma)/n. \tag{8}$$

The following result (proved in the appendix, alongside all other theoretical results in this work) will be exploited in the sequel to show that the use of synthetic data in model training can lead to catastrophic effects regarding test error.

**Theorem 1.** *Define $\sigma^2 := p_1 \sigma_1^2 + p_2 \sigma_2^2$ and let $\kappa, u \geq 0$ be as previously constructed. In the proportionate scaling limit (4), the test error w.r.t the true data distribution $P_1$, of the classical linear model $\widehat{f}_{CL}$ defined in (3) is given by $E_{test}(\widehat{f}_{CL}) \simeq \overline{E} + \zeta$, with*

$$\overline{E} = B + V, \quad V = \sigma^2 \frac{\mathrm{df}_2(\kappa; \Sigma)/n}{1 - \mathrm{df}_2(\kappa; \Sigma)/n}, \quad B = \kappa^2 \frac{\mathrm{tr}\, \Gamma\Sigma(\Sigma + \kappa I_d)^{-2}}{1 - \mathrm{df}_2(\kappa; \Sigma)/n},$$

$$\zeta = p_2^2 \cdot (1 + p_1 u) \, \mathrm{tr}\, \Delta\Sigma^3 (\Sigma + \kappa I_d)^{-2} + p_2 u \, \mathrm{tr}\, \Delta\Sigma(p_1 \Sigma + \kappa I_d)^2 (\Sigma + \kappa I_d)^{-2}.$$

Note that for $\Delta = 0$ (i.e $w_2^* = w_1^*$), which corresponds to assuming that the real data and the surrogate data have the same distribution, the above theorem gives $E_{test}(\widehat{f}_{CL}) \simeq \overline{E} \simeq B + V$ which is the classical bias-variance decomposition (Hastie et al., 2022; Richards et al., 2021) for ridge regression on $n$ samples from the distribution $P_{\Sigma,w_1^*,\sigma^2}$. The extra term $\zeta$ appearing in Theorem 1 is responsible for model collapse! In Appendix D.2, we show how Theorem 1 recovers the main results of Dohmatob et al. (2024a) for special choices of the displacement matrix $\Delta$.

**Strong Model Collapse.** In particular, in the "scaling laws" regime where $\phi \to 0^+$, it holds that $\zeta \simeq p_2^2 \operatorname{tr} \Delta$. In this case, if $\operatorname{tr} \Delta$ remains bounded away from zero, then so is $\zeta$ unless $p_2 \to 0^+$, i.e we discard all synthetic data from the training dataset. This is *strong* model collapse. It hints that model collapse as exposed by Shumailov et al. (2023; 2024); Hataya et al. (2023); Martínez et al. (2023a;b); Bohacek & Farid (2023); Briesch et al. (2023); Guo et al. (2023) cannot be fixed by naively mixing synthetic and real data during training. We show in Section 3.2 that this observation continues to hold in the setting of random projections model $\widehat{f}_{RP}$ defined in (5). Finally, in Section 5 we study what happens when the synthetic data and real data are strategically mixed during training.

**Proving Theorem 1.** It turns out that the analysis of the classical linear model's test error $E_{test}(\widehat{f}_{CL})$ in Theorem 1 amounts to the analysis of the trace of rational functions of sums of random matrices. Although the limiting spectral density of sums of random matrices is a classical computation using subordination techniques (Marčenko & Pastur, 1967; Kargin, 2015), this is not enough for the full analysis; a more involved analysis is required. For example, some of the quantities we must analyze are of the following form (where $M_j := X_j^\top X_j / n$, $M := M_1 + M_2$; $A$ and $B$ deterministic matrices): $r_j^{(3)}(A, B) := \mathbb{E} \operatorname{tr} A M_j (M + \lambda I_d)^{-1} B (M + \lambda I_d)^{-1} M_j$. The difficulty will be even greater in the setting of random projections $\widehat{f}_{RP}$ because it leads to more complicated terms. To the rescue, in Appendix E we shall employ operator-valued free probability theory (OVFPT) to compute the exact high-dimensional limits of quantities like the definition of $r_j^{(3)}(A, B)$ above . The tools of OVFPT have been used in the recent machine learning theory literature to obtain precise asymptotics for the test error of neural networks (trained on real data only) in various linearized settings (Adlam & Pennington, 2020; Tripuraneni et al., 2021; Lee et al., 2023). The idea is to construct a block matrix $Q$ each of whose blocks is a constant or is proportional $X_1$, $X_1^\top$, $X_2$, or $X_2^\top$, and one of the blocks $Q^{-1}[i, j]$ of $Q^{-1}$ is equal to the original matrix $W = A M_j (M + \lambda I_d)^{-1} B (M + \lambda I_d)^{-1} M_j$. Such a $Q$ is referred to as a *linear pencil* for $W$. Because of the structure of $Q$, OVPT allows us to compute the limiting value of the expectation of the traces of the square blocks of $Q^{-1}$, and we ultimately extract $r_j^{(3)}(A, B) \simeq \lim \mathbb{E} \operatorname{tr} Q^{-1}[i, j]$.

**Example: The Isotropic Case.** To help unpack Theorem 1, consider the following concrete setup

$$\Sigma = I_d, \quad \Gamma = (r^2/d)I_d, \quad \Delta = (c^2/d)I_d,$$

for some constants $r, c > 0$. The constant $c^2$ captures how close the distribution of the synthetic data $P_2$ is to the distribution of the real data $P_1$; thus it captures the quality of the synthetic data. This gives $u \simeq \phi/((1+\kappa)^2 - \phi)$, where $\kappa > 0$ uniquely satisfies the fixed-point equation $\kappa - \lambda = \kappa\phi/(1+\kappa)$; this is a quadratic equation that characterizes the well-known Marchenko-Pastur law (Marčenko & Pastur, 1967). The quantities appearing in the formulae presented in Theorem 1 then take the following simple forms: $V = \sigma^2\phi/((1+\kappa)^2 - \phi)$ and $B = \kappa^2 r^2/((1+\kappa)^2 - \phi)$, and

$$\zeta = \big(p_2(1 + p_1 u) + (p_1 + \kappa)^2 u\big) \, p_2 c^2/(1+\kappa)^2.$$

In particular, in the unregularized limit $\lambda \to 0^+$ corresponding to OLS, we get $\kappa \to (\phi - 1)_+$. To further make things concrete, consider the under-parametrized case where $\phi \in (0, 1)$ in the proportionate scaling regime (4). The over-parametrized case $\phi \in (1, \infty)$ is treated in Appendix D.1. We deduce the following corollary to Theorem 1.

**Corollary 1.** *Suppose $\phi \in (0, 1)$. Then, in the limit (4) and $\lambda \to 0^+$, the test error with respect to the true data distribution $P_1$, of the classical linear model $\widehat{f}_{CL}$ defined in (3) is given by*

$$E_{test}(\widehat{f}_{CL}) \simeq \sigma^2\phi/(1 - \phi) + \big(p_2^2 + p_2 p_1 \phi/(1 - \phi)\big) c^2.$$

*Moreover, for fixed $c > 0$ and small $\phi \in (0, 1)$, it holds that $E_{test}(\widehat{f}_{CL}) \simeq \sigma^2 d/n + p_2^2 c^2 + O(\phi^2)$. In particular, if $c^2 = \Omega(1)$, i.e bounded away from zero (corresponding to low-quality synthetic data), then $E_{test}(\widehat{f}_{CL}) = \Omega(p_2^2 c^2)$: the scaling law plateaus unless $p_2 \to 0^+$, i.e unless all but a vanishing proportion of synthetic data is discarded from the training dataset.*

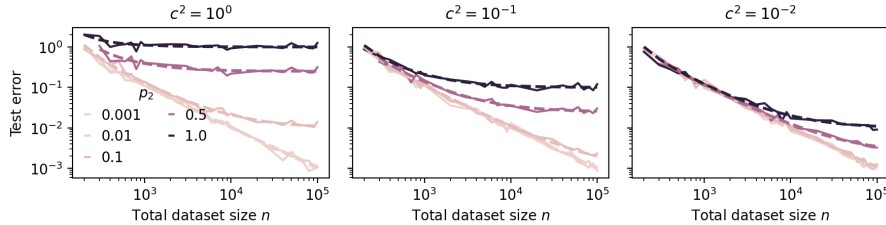

Figure 3: **Strong model collapse in the classical linear model** (empirical confirmation of Corollary 1). The training dataset comprises of $n = n_1 + n_2$ samples from a mixture of $n_2 = p_2 n$ synthetic samples and $n_1 = n - n_2$ real samples. The real samples are from the same distribution as the real / true samples of the training dataset, while the synthetic samples are from a distribution with the same covariance structure and label noise level $\sigma = 1$, but an incorrect labelling function (epistemic error). The quality of the synthetic data is controlled by the scalar $c$, with $c \to 0$ corresponding to synthetic data of perfect quality (higher values correspond to lower quality synthetic data). Solid curves correspond to experiments, and broken curves correspond to our theoretical predictions of Corollary 1; notice the perfect match. We see that even a small amount of low-quality synthetic data is enough to cause model collapse, whereby the test error of the model deviates from a perfect diagonal (ideal scaling law, corresponding to $p_2 = 0$, i.e training on real data only).

The result is empirically illustrated in Figure 3, where we see that even a small amount of low-quality synthetic data is enough to cause model collapse, whereby the test error of the model deviates from a perfect diagonal (ideal scaling law, corresponding to $p_2 = 0$, i.e training on real data only).

**Remark 2.** *Corollary 1 can be extended to the the non-isotropic case, but the statement is much longer and is thus omitted here.*

## 3.2 RANDOM PROJECTIONS MODEL

We now turn to the more challenging setting of the random projections model $\widehat{f}_{RP}$ given in (5). As mentioned before (cf. Section 2.2), such models are considered in our work as a simplification of the high-dimensional dynamics of actual neural networks, which still allows us to capture the effect of model size in the model collapse phenomenon.

We will need the following scalars which capture the high-dimensional statistics of the model $\widehat{f}_{RP}$.

**Definition 2.** *Let $(e, \tau, u, \omega)$ be the unique positive solution to the following fixed-point equations*

$$1/e = 1 + \psi\tau\bar{\mathrm{tr}}\,\Sigma K^{-1}, \quad 1/\tau = 1 + \bar{\mathrm{tr}}\,K_0 K^{-1}, \quad \text{with } K_0 := e\Sigma,\ K := \gamma\tau K_0 + \lambda I_d, \quad (9)$$

$$u = \psi e^2 \bar{\mathrm{tr}}\,\Sigma(\gamma\tau^2 L' + \omega I_d)K^{-2}, \quad \omega = \tau^2\bar{\mathrm{tr}}\,(\gamma\omega K_0^2 + \lambda^2 L')K^{-2}, \quad \text{with } L' := (1 + u)\Sigma. \quad (10)$$

*Here, $\bar{\mathrm{tr}}\,A := \mathrm{tr}\,A/d$ is the normalized trace. Also define $\theta := \lambda/(\gamma\tau e) > 0$, $\omega' := \omega/(\gamma\tau^2) > 0$.*

As usual, we denote $\sigma^2 := p_1\sigma_1^2 + p_2\sigma_2^2$. Also, for any $p \in [0, 1]$ define a $d \times d$ positive definite matrix $T(\theta; p) := p\Sigma + \theta I_d$, and $T(\theta) := T(\theta; p)|_{p=1} = \Sigma + \theta I_d$. The following result is a nontrivial extension of Theorem 1 to the case of random projections.

**Theorem 2.** *In the proportionate scaling limit (6), the test error w.r.t the true data distribution $P_1$, of the random projections model $\widehat{f}_{RP}$ defined in (5) is given by $E_{test}(\widehat{f}_{RP}) \simeq \overline{E} + \zeta$, with*

$$\overline{E} \simeq B + V, \text{ where } B = (1 + u)\theta^2\,\mathrm{tr}\,\Gamma\Sigma T(\theta)^{-2} + \omega'\,\mathrm{tr}\,\Gamma\Sigma^2 T(\theta)^{-2},$$

$$V = \left(\mathrm{tr}\,\Sigma^2 T(\theta)^{-2} + (\omega' - \theta u)\,\mathrm{tr}\,\Sigma T(\theta)^{-2}\right)\sigma^2/e, \quad (11)$$

$$\zeta = p_2(p_2 + p_1 u)\,\mathrm{tr}\,\Delta\Sigma^3 T(\theta)^{-2} + p_2(\omega' + 2p_1 u\theta)\,\mathrm{tr}\,\Delta\Sigma^2 T(\theta)^{-2} + p_2 u\theta^2\,\mathrm{tr}\,\Delta\Sigma T(\theta)^{-2}.$$

We now explore a few important consequences of Theorem 2.

**A Double-Descent Curve.** The bias-variance decomposition presented in Theorem 2 is empirically illustrated in Figures 2 and 4 for the Gaussian setting (1) (see Appendix B.1 and A.1 for details on the experimental setup and additional results in this setting). Notice the perfect match with experiments. The shape of the bias curve in Figure 2 (leftmost plot) is reminiscent of the well-known double-descent (Bach, 2023) in the unregularized setting $\lambda \to 0^+$. The divergence at the interpolation threshold $m = n$ (i.e. $\psi = 1$) is because the bias term $B$, the variance term $V$, and the extra term $\zeta$ (responsible for model collapse) all diverge to infinity at this point.

**Strong Model Collapse.** Observe that the first term in the expression for $\zeta$ given in Theorem 2 is lower-bounded by $p_2^2 \operatorname{tr} \Delta \Sigma^3 (\Sigma + \theta I_d)^{-2}$, which scales linearly with the square of the proportion $p_2 \simeq n_2/n$ of synthetic data in the training dataset $\mathcal{D}$. However, unless $p_2 \to 0^+$, i.e unless the proportion $p_2$ of synthetic data in the training dataset vanishes, the performance of the model eventually plateaus above the baseline $\overline{E}$ (corresponding to the setting where all training data is real, i.e no synthetic data in training dataset). This is strong model collapse.

Since the factor $\operatorname{tr} \Delta \Sigma_3 (\Sigma + \theta I_d)^{-2}$ only depends on the design choices of the model (via the scalar $\theta$ defined previously), we expect that different design choices (e.g., model size) will lead to different model collapse profiles.

**Are Larger Models More Prone or Less Prone to Model Collapse?** Figure 1 shows the results of a small experiment to investigate this. The input dimension is $d = 600$, and the covariance matrix is identity $I_d$ (isotropic features). The total number of training examples is fixed to $n = 500$. The $\Delta$ matrix is taken to be of the form $\Delta = (c^2/d)\Sigma^{-1}$ (similar results are observed for different covariance matrices) for different values of $c^2$ as follows: $c^2 = 0$ (synthetic data of very high quality), represented with square markers; $c^2 = 0.1$ (high quality synthetic data), represented with diamonds; $c^2 = 0.5$ (low quality), represented by triangles; and $c^2 = 1$ (very low-quality synthetic data), represented by stars. As indi-

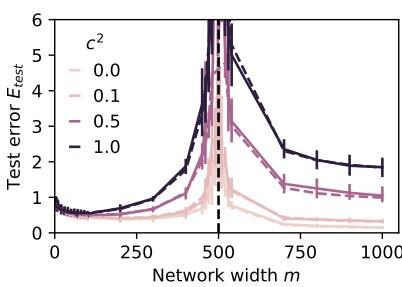

Figure 4: **Impact of model size (network width $m$) on model collapse**. As usual, solid curves correspond to experimental results (5 runs), while broken curves correspond to predictions of our theory (here, Corollary 4). Error bars correspond to 5 independent runs. Also see Figures 2 and 7.

cated on the figure, the leftmost plot corresponds to the regime where there is much fewer synthetic than real samples ($n_2 = 50$ synthetic samples versus $n_1 = 450$ real samples). Here, for both very high-quality and high-quality (squares and diamonds), the optimal tradeoff is struck by larger models (i.e, larger values of $\psi$). For lower-quality data (triangles and stars), the frontier shifts upwards and from left to right; **intermediately sized models become optimal for coping with model collapse**.

In the middle plot, size of the synthetic dataset is comparable to the size of the real dataset ($n_2 = 200$ versus $n_1 = 300$). For high-quality synthetic data, larger models are still better than smaller models. However, for this setting, the frontier shifts upwards and from left to right, and the optimal model size is intermediate. For the rightmost plot, the size of the synthetic dataset is considerably larger than the real dataset ($n_2 = 400$ versus $n_1 = 100$). The results are similar to the case $n_2 = 200$ except that the Pareto frontiers are higher over the diagonal (indicating more serious model collapse). In all cases, very small models are never optimal: they are not good even in the classical sense when training is done only on real data, and the presence of synthetic data only makes this worse.

**Special Cases of Theorem 2** In the limit $p_2 \to 0^+$ (i.e., no synthetic data; all the training data is real), $\zeta \to 0$ in Theorem 2, and we recover the main result of Bach (2023) as a special case, namely $E_{test}(\widehat{f}_{RP}) \simeq B + V$, with $B$ and $V$ as given in the theorem. Note that even in this special case, our result is more general since it covers the entire regularization path while the formulae in Bach (2023) are only for the unregularized case $\lambda \to 0^+$. On the other hand, Theorem 2 is a generalization of Theorem 1, as can be seen by taking $\psi \to \infty$. Refer to Appendix G.2 for details.

## 4 EXPERIMENTAL RESULTS

Our theoretical framework is developed within the context of high-dimensional linear regression and random projections models using Gaussian data. Our first departure from the confines of our theory are experiments with two-layer neural networks trained on the MNIST dataset (Deng, 2012) both in the random feature model (with ReLU activations) and with fully trained networks. These are presented in Appendix A.2. We find that the general trends observed in our asymptotic theory still hold: (1) there is significant model collapse, which only diminishes as the fraction of synthetic data approaches 0; (2) larger models exhibit a more severe model collapse (Figures 8 and 9).

We now provide evidence that our theory is applicable to large-scale problems, particularly in the context of language modeling with GPT-2 models. The BabiStories dataset (Zhang et al., 2024a), a reproduction of TinyStories (Eldan & Li, 2023) using the Mixtral-8x7B open language model (Jiang et al., 2024) enables us to study language modeling with relatively small models in a compute-

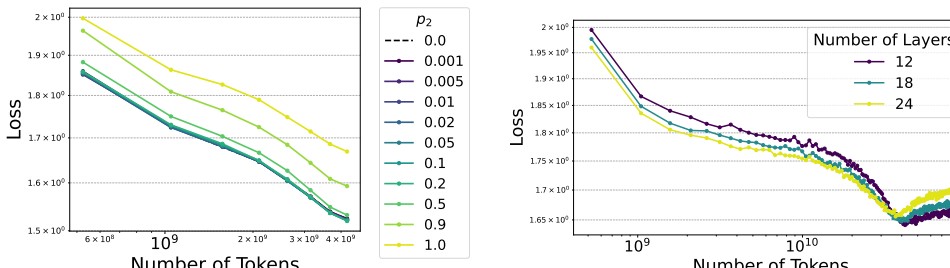

Figure 5: **Results on BabiStories with GPT-2 models.** Synthetic BabiStories is generated with a trained GPT-2-small with the same set of prompts. **(Left)** Impact of the proportion of synthetic data $p_2$ on model collapse in a language model with 12 layers. **(Right)** Impact of model size (number of layers) on model collapse. Here the model is trained on synthetic data only (i.e $p_2 = 1$). The loss is evaluated on the TinyStories test set.

efficient and environmentally friendly way. It comprises stories generated by prompting large models to create narratives in simple language that a three-year-old child could understand, effectively simplifying the underlying language model.

**Setup.** We train a GPT-2-small model with 124 million parameters on the BabiStories dataset as the generator. Using the same story prompts, which include the story beginning and character names, the generator creates our synthetic dataset. We then mix this synthetic dataset with the original BabiStories, train, and evaluate model perplexity on a validation set derived from the original BabiStories. Detailed information on optimization and model configurations is provided in Appendix B.3.

**Impact of Synthetic Data Proportion.** We investigate the effect of varying the synthetic data proportion ($p_2$) on the model's scaling in Figure 5 (left). Here, the x-axis represents the number of tokens used to train the model. In this experiment, the synthetic data is of high quality, as evidenced by the low training loss and coherent text generations, corresponding to the small $c^2$ (cf. Definition 1) case in our illustrative Figure 1. Consequently, even moderate amounts of synthetic data delay the progression of the scaling laws, and we expect this to eventually lead to plateaus or at least very bad bad (i.e small) exponents in the final scaling laws as predicted in Dohmatob et al. (2024b) in the special case of training on synthetic data only.

**Impact of Model Size.** We next examine the impact of model size on training with synthetic data. In addition to the GPT-2-small model (12 layers), we introduce two larger models: one with 18 layers (166 million parameters) and another with 24 layers (204 million parameters). The embedding dimension and number of attention heads remain constant across all models. We generate a synthetic dataset 10 times larger than the original one to support the scaling of tokens. As shown in Figure 5 (right), larger (deeper) models maintain a lower test loss until the dataset size increases—likely exceeding the interpolation threshold—at which point smaller models begin to exhibit lower loss and reduced overfitting. This aligns with the predictions of Theorem 2 (also refer to Figure 1, 2, and the discussion just after the theorem), which suggest that larger models tend to amplify model collapse beyond the interpolation threshold. In Figure 5, we observe this amplification when the number of tokens exceeds $3 \times 10^{10}$. Conversely, the theory predicts that over-parameterized models help mitigate collapse, a trend we observe when the number of tokens is below $1 \times 10^{10}$, leading to improved performance of larger models.

## 5 CAN STRATEGIC DATA MIXING SCHEMES PREVENT MODEL COLLAPSE?

Having established the occurrence of strong model collapse both theoretically and empirically, we now explore strategies to mitigate it and leverage synthetic data under stronger assumptions. We begin by assuming clear information about the data source and consider the following strategic iterative mixing, inspired by Ferbach et al. (2024). In this approach, a model is fitted on a mixture of synthetic and real data. In the next iteration, the labels for the synthetic data are replaced with the labels predicted by the previous iteration of the process, and so on.

For concreteness, take $\Sigma_1 = \Sigma_2 = \Sigma = I_d$ for the covariance matrices, and $\Delta = (c^2/d)\Sigma^{-1} = (c^2/d)I_d$. In this setup, the proposal of Ferbach et al. (2024) then becomes: At iteration $t + 1$, we mix $n_2 = p_2 n$ samples of synthetic data from a source having quality parameter $c^2 = c_t^2$, with $n_1 = n - n_2$ samples of real data to construct a penalized linear model $\widehat{w}^{(t+1)}$ according to (3). This trained model generates the synthetic data with $c^2 = c_{t+1}^2$.

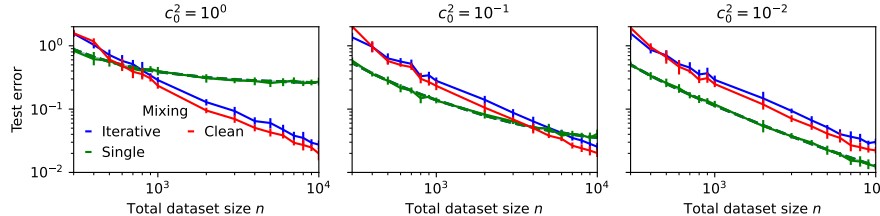

Figure 6: **Iterative vs Single-Step Mixing.** Solid lines represent the experimental results (5 runs), while dashed lines correspond to the theoretical predictions of Corollary 2. The iterative mixing is repeated 5 times, with $p_1 = p_2 = 0.5$. "Clean" refers to the scaling when using solely the $n_1 = p_1 n$ real data in the dataset.

Thus, the idea is to iteratively enhance the quality of the synthetic data through bootstrapping.

**Corollary 2.** *For large $t$, it holds in the limit* (4) *and then $\phi, \lambda \to 0^+$ that* $E_{test}(\widehat{f}_{CL}^{(t)}) \simeq \overline{E}/(1 - p_2^2) + \Theta(p_2^{2t})$, *where* $\overline{E} \simeq \sigma^2 d/n$, $\quad \sigma^2 := p_1 \sigma_1^2 + p_2 \sigma_2^2$.

Let us explore some important consequences of Corollary 2.

**– Iterative Mixing Recovers Scaling Laws but Might Not be Feasible in Practice.** If the practitioner can curate a sufficient amount of data from the original distribution, the training dataset will include a non-vanishing proportion of real data, ensuring that $p_1$ remains bounded away from zero. By comparing $\overline{E}$ with $p_2^{2t}$, we observe that iterative mixing over $t$ iterations, where $t$ is of the order of $\log(n/d)$, results in a scaling law proportional to $\overline{E}$, as empirically confirmed in Figure 6. However, this comes at the cost of significant bootstrapping, a large volume of real data, and the need to clearly distinguish between real and synthetic data across iterations—conditions that are all too computationally expensive and challenging to implement in practice.

**– Iterative Mixing Relies Mostly on Real Data.** In Figure 6, we compare the scaling of iterative mixing with the scaling obtained using only the $p_1 n$ real data portion from the same training set ("Clean"). While the scaling rate remains consistent, iterative mixing consistently underperforms compared to using real data alone. This suggests that iterative mixing may primarily neutralize the synthetic data and heavily rely on the real data to recover the scaling. Even when the original synthetic data is of high quality (i.e., when $c_0$ is small, rightmost plot of FIgure 6), the iterative method fails to effectively leverage the synthetic data, resulting in worse performance than single mixing. Thus, although iterative mixing recovers the same scaling rate, the model still collapses to some degree, and no significant performance improvement is observed.

**– Iterative Mixing with Little Real Data is Bad.** If we consider the setting where we only have limited real data or where there is faster accumulation of synthetic data, which corresponds to $p_2 \to 1$ (the real data in the training set is diminishing), then it holds that for any $t \geq 1$, $E_{test}(\widehat{w}^{(t)}) \simeq c_0^2 + t\overline{E}$. This is an increasing function of $t$, meaning that there is still catastrophic model collapse.

## 6 DISCUSSION

Our work systematically characterizes the effects of training models on mixtures of real and synthetic data, showing that model collapse is a robust phenomenon that persists even with small fractions of synthetic data, in the asymptotic regime. By introducing new mathematical tools, we extend prior work to analyze more complex mixing settings and models (random projections), broadening the scope of theoretically tractable problems. Experiments confirm our theoretical predictions across large language models (LLMs) and also fully-trained feed-forward neural networks.

Going beyond the prevalent "neural scaling laws" paradigm (Kaplan et al., 2020; Hoffmann et al., 2022) which is at the basis of the current trend in training LLMs, this study emphasizes the importance of preserving and labeling real data, either by curating it or avoiding unintended synthetic data in training, reflecting a shift as AI-generated data becomes prevalent. Our work also delineates the impact of model size on the model collapse profile. Future work will explore the effect of other model design choices like activation functions, depth, and optimization hyper-parameters like learning rate and momentum. To this end, we can leverage "Gaussian equivalents" (Goldt et al., 2022) to extend our theory to wide, fully-trained networks in the neural tangent kernel (Jacot et al., 2018) and lazy (Chizat et al., 2019) regimes, using operator-valued free probability theory (Mingo & Speicher, 2017), like we have done in our analysis.

## 7 ACKNOWLEDGEMENTS

YF and JK acknowledge support through the NSF NRT training grant award 1922658. Part of this work was done while YF and JK where visiting the Centre Sciences de Donnees (CSD) at the Ecole Normale Superieure in Paris, France, and YF and JK wish to thank the CSD for their hospitality. YF would like to thank Jianyu Zhang for his help with the experiments involving GPT-2 on BabiStories.

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

# Appendix

## Table of Contents

# A FURTHER EXPERIMENTAL RESULTS

## A.1 ADDITIONAL RESULTS FOR THE TOY SETTING OF MULTIVARIATE GAUSSIANS

Figure 7 provides additional plots for various data quality parameters $c^2$ showing model collapse as a function of model size in the toy setting of multivariate Gaussians with random projections (experimental details in Section B.1).

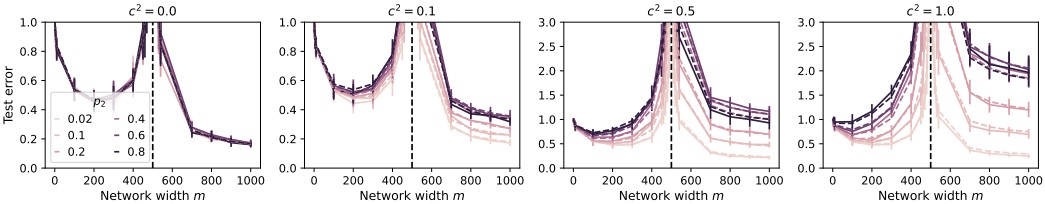

Figure 7: **Impact of model size (network width $m$) on model collapse.** Same setting as for Figure 4, but with quality parameter $c^2$ (smaller is better) as shown on top of each plot and proportion of synthetic data $p_2$ as in the legend (Figure 4 showed the reverse).

## A.2 EXPERIMENTAL RESULTS FOR NEURAL NETWORKS ON MNIST

**Setup.** For two-layer neural networks, we consider two scenarios: (1) learning with a random projection model as in Section 3.2, where the first layer of the network is fixed randomly, and only the second layer is trained, and (2) learning with a fully trainable neural network. The first setting directly corresponds to our theoretical results from Section 3.2, but with ReLU activation functions. In the case of fully trained neural networks in the second setting, our theory does not apply directly. However, we hypothesize that the general trends observed in our asymptotic theory will still hold: (1) there will be a significant model collapse, which only diminishes as the fraction of synthetic data approaches 0; (2) larger models will exhibit a more severe model collapse.

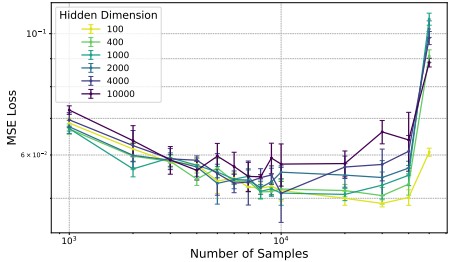

Figure 8: **Fully trained two-layer network on MNIST data.** Impact of model size (hidden dimension, aka network width) on model collapse. Here, the model is trained solely on synthetic data (i.e $p_2 \to 1$).

To align with the theoretical setting, we employ a (multivariate) regression approach where labels are converted to one-hot vectors and the model is trained using mean squared error. The synthetic labels were generated by another two-layer network, with Gaussian label noise (standard deviation of 0.1) added. A validation set is used to select the best checkpoint, and evaluation is conducted on the test set using the clean labels. Further details of the training are provided in Appendix B.2.

**Results.** Figure 9 presents the results for both random feature models (left) and fully trained neural networks (right). In these experiments, we mixed synthetic and original data in the training set with varying coefficients, $p_1$. As the proportion of synthetic data, $p_2$, increases, the scaling laws slow down and eventually plateau. We observe a strong model collapse: only when $p_2$ approaches 0 does the collapse subside. The results are consistent across both cases, validating our theoretical predictions and demonstrating the applicability of our insights to more complex scenarios.

We also investigated how model size, specifically the hidden dimension of fully trained neural networks, affects model collapse. As shown in Figure 8, models with varying hidden dimensions were trained exclusively on the synthetic dataset with $p_2 = 1$. For training sets ranging from 10,000 to 50,000 samples, our results indicate that larger models are more susceptible to model collapse under the same validation and evaluation protocols. Notably, all these models remain in the interpolation regime, aligning with our theoretical predictions.

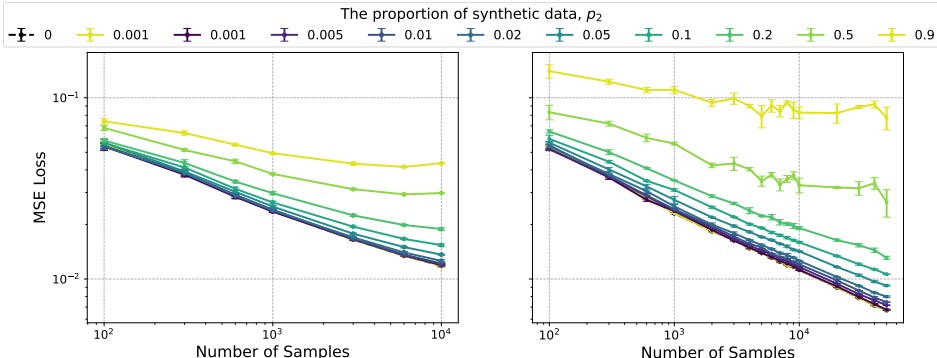

Figure 9: **Model collapse as a function of the proportion of synthetic data.** We use the MNIST dataset with regression loss. Error bars correspond to 5 runs. **Left**, Random feature model with hidden dimension 100,000. **Right**, Two-layer neural network of width (i.e hidden dim.) $m = 2000$.

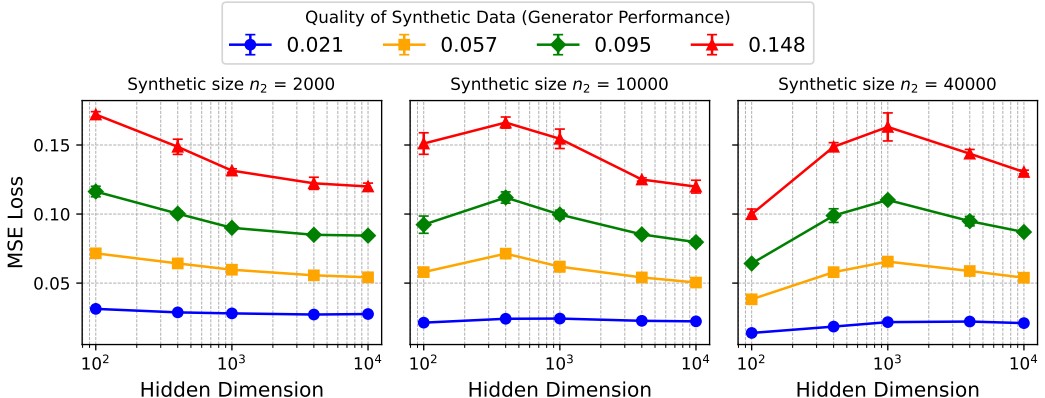

Figure 10: **Understanding the role of model size in model collapse under varying qualities of synthetic data and dataset sizes.** The quality of the synthetic data is evaluated using the MSE loss on the test set. The model is trained solely on synthetic data ($p_2 \to 1$).

### A.3 GENERAL PICTURE FOR NEURAL NETWORK ON MNIST

To provide a comprehensive understanding of how the quality of synthetic data, the quantity of synthetic data, and the network size impact performance, we conducted a large-scale experiment varying these factors, as shown in Figures 10 and 11. Figure 10 uses the MSE loss as an indicator of synthetic data quality, while Figure 11 uses accuracy as the indicator. To simplify the analysis, we focus on pure synthetic data ($p_2 = 1$).

The synthetic data with the highest quality already achieves accuracy close to the optimal. As the quality decreases, we observe that the shape of the curve begins to resemble a double descent curve, similar to the changes in the Pareto frontiers shown in Figure 1. With different combinations of the number of synthetic data $n_2$ and hidden dimension $d$, the figure captures various segments of the double descent curve depicted in Figure 4. When $n_2$ is small (as seen in the left subplots), it corresponds to a large parameterization rate $\psi$, placing it in the second descent region of the double descent curve. Conversely, when $n_2$ is large (as shown in the right subplots), it captures the up-and-down behavior characteristic of the double descent phenomenon.

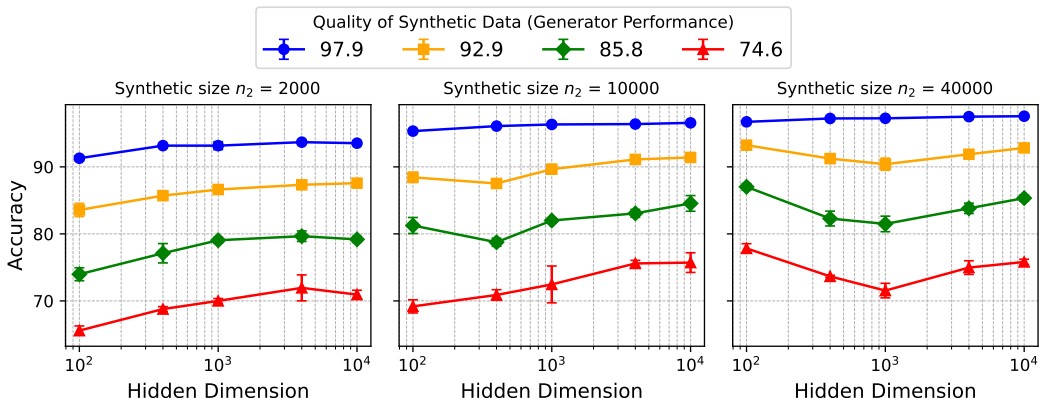

Figure 11: **Understanding the role of model size in model collapse under varying qualities of synthetic data and dataset sizes.** The quality of the synthetic data is evaluated using the accuracy on the test set. The model is trained solely on synthetic data ($p_2 \to 1$).

## B  EXPERIMENTAL DETAILS

### B.1  TOY SETTING: RANDOM PROJECTIONS MODEL

**Setup.** As a sanity check to empirical confirm our analytical predictions from Theorem 2, we consider a setting with multivariate Gaussian data (1). The feature covariance matrix $\Sigma$ is constructed to have power-law eigenvalues $\lambda_j = C/j$, where $C$ is such that $\operatorname{tr}\Sigma = \lambda_1 + \ldots + \lambda_d = 1$. The ground-truth labelling weights $w_1^*$ of the real data distribution $P_1$ sampled from $N(0, (1/d)I_d)$, while the ground-truth weights $w_2^*$ for the synthtic data distribution are sampled from $N(w_1^*, \Delta)$ with $\Delta = (c^2/d)\Sigma^{-1}$ for different values of $c^2$ ranging from $\{0, 0.1, 0.5, 1\}$ which controls for the quality of the synthetic data. We run a small experiment with label noise levels $\sigma_1 = \sigma_2 = 0.1$, input-dimension $d = 600$, number of real samples $n_1 = 300$, and synthetic samples $n_2 = 200$, for a total of $n = n_1 + n_2 = 500$ samples. We fit a random projection model $\widehat{f}_{RP}$ according to (5) and for different values of the width parameter $m$ (to control the size the of the model), and report the results in Figures 4 and 7. The regularization parameter $\lambda$ is set to a very small value ($10^{-8}$). We also consider a variation of this experiment with different values of the synthetic dataset size $n_2$ and report the results in Figure 1.

### B.2  TWO-LAYER NEURAL NETWORKS

The two-layer neural networks are trained using stochastic gradient descent (SGD) with a batch size of 128 and a learning rate of 0.1. The models are trained for 400 epochs to fully converge. We employ a held-out validation set from the training set to select the best checkpoint to evaluate.

### B.3  LANGUAGE MODELING

The generation process for the BabiStories dataset is detailed in the GitHub repository of Zhang et al. (2024a). The dataset comprises a training set of 2,200,000 stories and a validation set of 22,000 stories, created by prompting the Mistral-8x7B model. Each prompt includes a description of the generation task, character names, specific words to be used, and the beginning of a story. The dataset stores the beginnings of these stories along with their generated continuations.

In our experiments, we trained a GPT-2-small model on this dataset to generate synthetic data. The model was trained using next-token prediction, utilizing the beginnings and continuations of stories to have good story generation quality. To maintain consistency with the original prompt distribution, we used all the prompts that were initially employed to generate BabiStories. During story generation, we applied a temperature setting of 1 with top-p decoding where $p = 1$. After generation, we filtered out stories of poor quality, such as those containing unwanted symbols, following the same

procedure as in Zhang et al. (2024a). The filtered beginnings and synthetic continuations were then collected to form the synthetic dataset.

We used a GPT-2 model with an embedding dimension of $d = 768$, 12 attention heads, and a context length of 512 tokens, which typically encompasses one to three stories. During training, we applied a learning rate of $5 \times 10^{-3}$, a dropout rate of 0.05, L2 weight decay of 0.1, and a warm-up phase of 2,000 iterations.

## C STATIC / SINGLE-STEP DATA MIXING

For the purposes of studying scaling laws for learning on mixtures of real and surrogate (e.g synthetic data), the setting considered in Jain et al. (2024) consists in the following optimization problem:

$$\widehat{w} = \arg \min_{w \in \mathbb{R}^n} \frac{1-\alpha}{n_1} \sum_{(x_i, y_i) \in \mathcal{D}_1} (x_i^\top w - y_i)^2 + \frac{\alpha}{n_2} \sum_{(x_i, y_i) \in \mathcal{D}_2} (x_i^\top w - y_i)^2 + \lambda \|w\|^2. \quad (12)$$

This is an instance of weighted weighted empirical risk minimization (Shimodaira, 2000; Vogel et al., 2021) where the weight the sample weight $\pi_i$ is constant across each group: $\pi_i = (1 - \alpha)n/n_1 \simeq (1-\alpha)/p_1$ for real samples real samples vs $\pi_i = \alpha n/n_2 \simeq \alpha/p_2$ for synthetic samples. Thus $\alpha \in (0,1)$ is a mixing coefficient for the two the two source of data; in particular $\alpha \to 0$ corresponds to only using real data (which corresponds to group 1) for training, while $\alpha \to 1$ corresponds to only using surrogate data (group 2). Formula (12) replaces the formula for the weights vector $\widehat{w}$ of the classical linear model $\widehat{f}_{CL}$ (3).

For conciseness, as in Section 5 we focus on the isotropic case considered in Section 3.1 where the feature covariance matrices are $\Sigma_1 = \Sigma_2 = I_d$ and the shift matrix $\Delta := \mathrm{cov}(w_1^* - w_2^*)$ has the form $(c^2/d)I_d$ for some scalar $c > 0$. Further, let us consider the regime where $d/n \to 0$. In the language of our paper, one should think of this as corresponding to the proportionate scaling regime given in (4), and then letting $\phi \to 0^+$ (extremely under-parametrized regime). We have the following result.

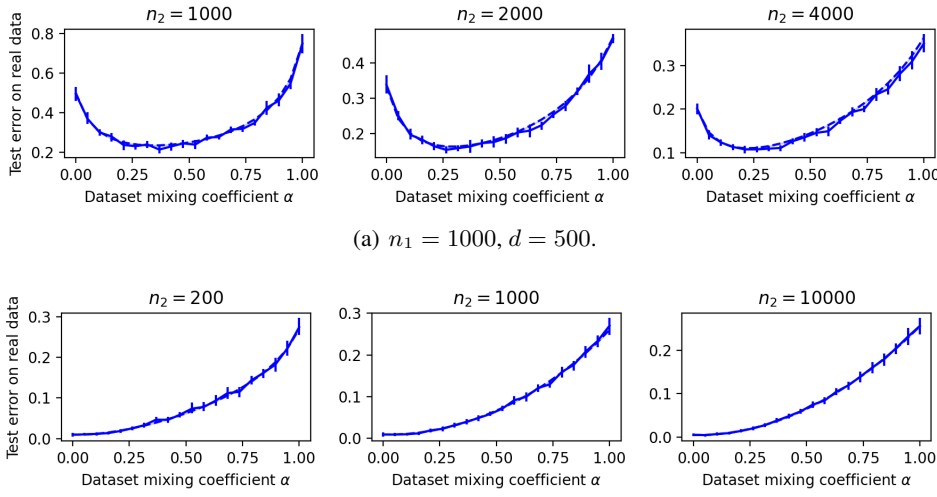

(a) $n_1 = 1000$, $d = 500$.

(b) $n_1 = 10000$, $d = 100$, so that $\phi = d/n \leq 100/10200 < 0.01$ (small). Corollary 3 correctly predicts that the optimal strategy mixing coefficient is $\alpha_* \approx 0$, i.e to discard surrogate data altogether.

Figure 12: **Failure of naive real+surrogate data mixing to solve model collapse**. For this experiment, we use different several different values for the size of the real data $n_1$ and the synthetic data $n_2$. Solid curves correspond to experiments while broken curves correspond to our theoretical prediction give in Corollary 3. Error-bars correspond to independent runs.

**Corollary 3.** *Consider the proportionate scaling limit (4). For small $\phi = d/n$, it holds that*

$$E_{test}(\widehat{f}_{CL}) \simeq p_2^2 \alpha^2 \, \mathrm{tr} \, \Delta + ((1-\alpha)p_1\sigma_1^2 + \alpha p_2\sigma_2^2)\phi + O(\phi^2). \quad (13)$$

The formula given in (13) represents a U-shaped function of $\alpha$, minimized when $\alpha = \alpha_*$, with

$$\alpha_* = \text{clip}_{[0,1]} \left( 1 - \frac{p_1 \sigma_1^2 - p_2 \sigma_2^2}{2 \, \text{tr} \, \Delta} \phi \right). \tag{14}$$

It should be clear that if $\text{tr} \, \Delta = \Omega(1)$ and $\sigma_1, \sigma_2 = O(1)$, then $\alpha_* \to 0$; this corresponds to only using real data for training! In contrast any fixed value $\alpha \in [0, 1)$, leads a positive lower-bound on test error $E_{test}(\widehat{f}_{CL}) \gtrsim \text{tr} \, \Delta$; this is effectively model collapse. The situation is empirically confirmed in Figure 12.

# D  SOME OMITTED THEORETICAL RESULTS AND COMMENTS

## D.1  CLASSICAL LINEAR MODEL IN OVER-PARAMETRIZED ISOTROPIC SETTING

We now complement the analysis presented at the end of Section 3.1 with an analysis for the case $\phi \in (1, \infty)$. Plugging into Theorem 1 gives $\kappa \to \phi - 1$ and $u \to 1/(1 - \phi/\phi^2) = \phi/(\phi - 1)$ in the limit $\lambda \to 0^+$. We obtain the following corollary.

**Corollary 4.** *For $\phi \in (1, \infty)$, in the limit (4) and $\lambda \to 0^+$, it holds that $E_{test} \simeq \overline{E} + \zeta$, with*

$$\overline{E} = V + B, \; B = r^2 (1 - \frac{1}{\phi}), \; V = \frac{\sigma^2}{\phi - 1}, \quad \zeta = \frac{p_2 \, c^2}{\phi^2} \left( p_2 \frac{\phi - p_2}{\phi - 1} + (\phi - p_2)^2 \right), \tag{15}$$

*Moreover, for large $\phi \in (1, \infty)$, it holds that $E_{test}(\widehat{f}_{CL}) - \overline{E} \simeq (1 - 2/\phi) \, p_2 c^2 + O(1/\phi^2)$.*

Thus, for any fixed $c > 0$, strong model collapse occurs: the RHS vanishes only if $p_2 \to 0^+$, i.e only if we discard all but a vanishing proportion of synthetic data from the training dataset. This is strong model collapse. Combining with Corollary 1, we conclude that (at least in the isotropic setting, strong model collapse occurs both the under-parametrized and over-parametrized settings.

## D.2  CONNECTIONS TO CLASSICAL MODEL COLLAPSE IN REGRESSION

In the setting of classical model collapse (Shumailov et al., 2023; 2024; Alemohammad et al., 2023; Dohmatob et al., 2024b;a), we have $w_2^* = w_1^* + \sum_{\ell=1}^{N} X_\ell^\dagger E_\ell$, where $N$ is the number of iterations (i.e self-loops) in the synthetic data-generation process. Let $n_\ell$ be the number of samples available for training at stage $\ell$ with training data $(X_\ell, X_\ell w_2^* + E_\ell) \in \mathbb{R}^{n \times d} \times \mathbb{R}^n$, where the noise vectors $E_\ell$ are independent with iid components from $N(0, \sigma_\ell^2)$. In the proportionate scaling regime $n_1, \ldots, n_N \to \infty$ with $d/n_\ell \to \phi_\ell \in (0, \infty)$ for all $\ell$, the situation is equivalent to taking

$$\Delta = \sum_\ell \sigma_\ell^2 \cdot \mathbb{E} (X_\ell^\dagger)^\top X_\ell^\dagger \simeq \sum_\ell \frac{\sigma_\ell^2}{n_\ell - \text{df}_2(\kappa_\ell; \Sigma)} \Sigma (\Sigma + \kappa_\ell I_d)^{-2}, \; \text{with } \kappa_\ell = \kappa(n_\ell, 0; \Sigma).$$

In particular, if $\max_\ell \phi_\ell \leq 1$ (so that there is enough samples to perfectly fit the training data at each stage of the iterative process), and for simplicity we set $\sigma_\ell = \sigma_0$ for all $\ell$, then the above expression simplifies to $\Delta \simeq \left( \sum_\ell \sigma_\ell^2/(n_\ell - d) \right) \Sigma^{-1}$. More generally, consider the generic setting where $\Delta \simeq (c^2/d) \Sigma^{-1}$, for any $c > 0$, so that the previous setting corresponds to $c^2 = \sum_\ell \sigma_\ell^2 \phi_\ell/(1 - \phi_\ell)$.

In the particular case where $p_1 \to 0^+$, i.e only synthetic data is available for training. Theorem 1 then gives

$$\zeta \simeq \frac{c^2}{d} \cdot \left( \text{df}_2 + u \kappa^2 \, \text{tr}(\Sigma + \kappa I_d)^{-2} \right) = \eta^2 \, \text{df}_2 \cdot \left( 1 + \kappa^2 \frac{\text{df}_2}{n - \text{df}_2} \text{tr}(\Sigma + \kappa I_d)^{-2} \right).$$

In particular, taking $c^2 = \sum_\ell \sigma_\ell^2 \phi_\ell/(1 - \phi_\ell)$ gives

$$\zeta \simeq \left( 1 + \kappa^2 \frac{\text{df}_2}{n - \text{df}_2} \text{tr}(\Sigma + \kappa I_d)^{-2} \right) \frac{\text{df}_2}{d} \sum_\ell \frac{\sigma_\ell^2 \phi_\ell}{1 - \phi_\ell}. \tag{16}$$

This is recovers the main result of Dohmatob et al. (2024a).

# E  DETERMINISTIC EQUIVALENTS

Let $X_j$ (resp. $Y_j$) be the design matrix (resp. response vector) corresponding to dataset $\mathcal{D}_j$. Thus, the design matrix $X_1 \in \mathbb{R}^{n_1 \times d}$ for the real dataset has rows given by $x_i$ for $i \in [n_1]$ and $Y_1 \in \mathbb{R}^{n_1}$ with components $y_i$ for $i \in [n_1]$, with $X_2 \in \mathbb{R}^{n_2 \times d}$ and $Y_2 \in \mathbb{R}^{n_2}$ defined analogously for the synthetic dataset. Let $X \in \mathbb{R}^{n \times d}$(resp. $Y \in \mathbb{R}^n$) be the design matrix (resp. response vector) corresponding to the total dataset. We temporarily drop the condition $\Sigma_1 = \Sigma_2 = \Sigma$, and instead consider generally different covariance matrices $\Sigma_1$ and $\Sigma_2$ for the marginal distribution of the features $x$ under the real data distribution $P_1$ and the synthetic data distribution $P_2$.

## E.1  CLASSICAL LINEAR MODEL

Note that the weights $\widehat{w}$ of the model $\widehat{f}_{CL}$ given in (3) can be written explicitly as $\widehat{w} = RX^\top Y$, where $R := (X^\top X + n\lambda I_d)^{-1} = (X_1^\top X_1 + X_2^\top X_2 + n\lambda I_d)^{-1}$, a random matrix. Its test error $E_{test}(\widehat{f}_{CL})$ writes $E_{test}(\widehat{f}_{CL}) = \mathbb{E}_{X,Y}[(\widehat{f}_{CL}(x) - x^\top w_1^*)^2] = \mathbb{E}_{X,Y}\|\widehat{w} - w_1^*\|_{\Sigma_1}^2$. In Proposition 4, we shall show that the RHS in the above can be decomposed into a sum of simply random quantities of the form $r_j^{(k)}(A)$ and $r_j^{(k)}(A, B)$ that we now describe and analyze.

Let $A$ and $B$ be $d \times d$ positive-definite matrices with well-behaved spectral (this will be made precise latter) and let $\lambda > 0$. In analyzing the bias-variance decomposition of the test error, we are ultimately led to consider the following quantities

$$r_j^{(1)}(A) := \mathbb{E} \operatorname{tr} AM_j(M + \lambda I_d)^{-1}, \tag{17}$$

$$r^{(2)}(A, B) := \mathbb{E} \operatorname{tr} A(M + \lambda I_d)^{-1}B(M + \lambda I_d)^{-1}, \tag{18}$$

$$r_j^{(3)}(A, B) := \mathbb{E} \operatorname{tr} AM_j(M + \lambda I_d)^{-1}B(M + \lambda I_d)^{-1}M_j, \tag{19}$$

$$r_j^{(4)}(A, B) := \mathbb{E} \operatorname{tr} AM_j(M + \lambda I_d)^{-1}B(M + \lambda I_d)^{-1}, \tag{20}$$

where we recall that $M := M_1 + M_2$ and $M_j := X_j^\top X_j/n$.

Let $(e_1, e_2)$ be the unique negative solution to the following pair of fixed-point equations

$$e_1 = \frac{1}{1 + \phi \overline{\operatorname{tr}} \Sigma_1 K^{-1}}, \;\; e_2 = \frac{1}{1 + \phi \overline{\operatorname{tr}} \Sigma_2 K^{-1}}, \quad \text{with } K := p_1 e_1 \Sigma_1 + p_2 e_2 \Sigma_2 + \lambda I_d. \tag{21}$$

Also, define $(u_1, u_2)$ to be the unique positive solution to the pair of fixed-point equations

$$u_1 = \phi e_1^2 \overline{\operatorname{tr}} \Sigma_1 L' K^{-2}, \;\; u_2 = \phi e_2^2 \overline{\operatorname{tr}} \Sigma_2 L' K^{-2}, \quad \text{with } L' := p_1 u_1 \Sigma_1 + p_2 u_2 \Sigma_2 + \lambda B. \tag{22}$$

Consider the following deterministic matrices

$$
\begin{aligned}
C_j &:= p_j e_j^2 (B + p_{j'} u_{j'} \Sigma_{j'})\Sigma_1 + u_1 (p_{j'} e_{j'} \Sigma_{j'} + \lambda I_d)^2, \\
D_j &:= e_j B - \lambda u_j I_d + p_{j'}(e_j u_{j'} - e_{j'} u_j)\Sigma_{j'},
\end{aligned}
\tag{23}
$$

where $1' := 2$ and $2' = 1$.

The following will be crucial for proving Theorem 1 and its corollaries.

**Proposition 1.** *In the proportionate scaling limit (4), it holds for $j = 1, 2$ that*

$$r_j^{(1)}(A) \simeq p_j e_j \operatorname{tr} A\Sigma_j K^{-1}, \tag{24}$$

$$r^{(2)}(A, B) \simeq \operatorname{tr} AL'K^{-2}, \tag{25}$$

$$r_j^{(3)}(A, B) \simeq p_j \operatorname{tr} A\Sigma_j C_j K^{-2}, \tag{26}$$

$$r_j^{(4)}(A, B) \simeq p_j \operatorname{tr} A\Sigma_j D_j K^{-2}. \tag{27}$$

### E.2 RANDOM PROJECTIONS

For $d \times d$ deterministic matrices $A$ and $B$, define the following quenched quantities

$$r_j^{(1)}(A) := \mathbb{E} \operatorname{tr} ASRS^\top M_j, \quad r_j^{(3)}(A, B) := \mathbb{E} \operatorname{tr} AM_j SR^\top SBSRS^\top M_j,$$

$$r_j^{(4)}(A, B) := \mathbb{E} \operatorname{tr} AM_j SR^\top SBSRS^\top, \quad r^{(5)}(A, B) := \mathbb{E} \operatorname{tr} AM_1 SR^\top SBSRS^\top M_2, \tag{28}$$

where we recall that $R := (S^\top MS + \lambda I_m)^{-1}$, $M := M_1 + M_2$, $M_j := X_j^\top X_j / n$. These quantities will be useful because we may write

$$V_k = \sum_j \sigma_j^2 \frac{1}{n} \mathbb{E} \operatorname{tr} M_j SRS^\top \Sigma_k SRS^\top = \sum_{j=1}^{2} \frac{\sigma_j^2}{n} r_j^{(4)}(I_d, \Sigma_k),$$

$$B_k = \operatorname{tr} \Gamma \Sigma_k + \mathbb{E} \operatorname{tr} \Gamma M SRS^\top \Sigma SRS^\top M - 2 \operatorname{tr} \Gamma \Sigma_k SRS^\top M + \operatorname{tr} \Delta M_2 SRS^\top \Sigma_k SRS^\top M_2$$

$$= \operatorname{tr} \Gamma \Sigma_k + 2 r^{(5)}(\Gamma, \Sigma_k) + r_1^{(3)}(\Gamma, \Sigma_k) + r_2^{(3)}(\Gamma, \Sigma_k) - 2 r_1^{(1)}(\Gamma \Sigma_k) - 2 r_2^{(1)}(\Gamma \Sigma_k) + r_2^{(3)}(\Delta, \Sigma_k).$$

Each term in the above decomposition can now be analyzed via operator-valued free-probability theory. The following proposition will be heavily exploited in the prove of Theorem 2.

**Proposition 2.** *In the proportionate scaling limit* (6)*, it holds that*

$$r_j^{(1)}(A) \simeq p_j \gamma \tau e_j \operatorname{tr} A\Sigma K^{-1}, \quad r_j^{(3)}(A, \Sigma) \simeq p_j \operatorname{tr} A\Sigma C_j K^{-2},$$

$$r_j^{(4)}(A, \Sigma) \simeq p_j \gamma \operatorname{tr} A\Sigma DK^{-2}, \quad r^{(5)}(A, \Sigma) \simeq p_1 p_2 \gamma \operatorname{tr} A\Sigma^2 EK^{-2}, \tag{29}$$

*where the constants $e_1$ and $e_2$ and the matrices $C_1$, $C_2$, $D$, and $E$ are as in Theorem 2.*

### E.3 PROOF OF PROPOSITION 1

WLOG, we only consider the case $j = 1$, and suppress this subscript henceforth from all the $r_j^{(k)}$'s.

**Computing $r_j^{(1)}$.** We only do $j = 1$ as $j = 2$ is completely analogous. One can obtain a minimal $9 \times 9$ linear pencil $Q$ for the random matrix $R = AM_1(M + \lambda I_d)^{-1}$ such that $Q$ is a $9 \times 9$ block matrix (not shown here[1]) and $R = Q^{-1}[1, 5]/\lambda$ (using zero-based indexing). It follows that in the asymptotic limit, one has

$$r^{(1)}/d = \mathbb{E} \bar{\operatorname{tr}} R \simeq G_{1,5}, \tag{30}$$

where $G = (\operatorname{id} \otimes \mathbb{E} \bar{\operatorname{tr}})[Q^{-1}] \in M_9(\mathbb{C})$ is the matrix containing the limiting expected values of the normalized traces of the blocks of each of the $9 \times 9 = 81$ blocks of $Q^{-1}$ (we define the trace of each rectangular as zero). Using classical operator-valued free probability theory (OVFPT) (Mingo & Speicher, 2017), we have the following fixed-point equations which define $G_{1,5}$ implicitly

$$G_{1,5} = p_1 G_{3,3} \bar{\operatorname{tr}} A\Sigma_1 (\lambda I_d + p_1 G_{3,3} \Sigma_1 + p_2 G_{7,7} \Sigma_2)^{-1}, \tag{31}$$

$$G_{3,3} = \frac{\lambda}{\lambda - \phi G_{4,2}}, \tag{32}$$

$$G_{7,7} = \frac{\lambda}{\lambda - \phi G_{8,6}}, \tag{33}$$

$$G_{4,2} = -\lambda \bar{\operatorname{tr}} \Sigma_1 (\lambda I_d + p_1 G_{3,3} \Sigma_1 + p_2 G_{7,7} \Sigma_2)^{-1}, \tag{34}$$

$$G_{8,6} = -\lambda \bar{\operatorname{tr}} \Sigma_2 (\lambda I_d + p_1 G_{3,3} \Sigma_1 + p_2 G_{7,7} \Sigma_2)^{-1}. \tag{35}$$

We deduce that $G_{3,3} = e_1$, $G_{7,7} = e_2$, and

$$r^{(1)}/d = G_{1,5} = p_1 e_1 \bar{\operatorname{tr}} A\Sigma_1 (\lambda I_d + p_1 e_1 \Sigma_1 + p_2 e_2 \Sigma_2)^{-1},$$

where $(e_1, e_2)$ is the unique pair of nonnegative solutions to the system of equations

$$e_1 = \frac{1}{1 + \phi \bar{\operatorname{tr}} \Sigma_1 (\lambda I_d + p_1 e_1 \Sigma_1 + p_2 e_2 \Sigma_2)^{-1}}, \tag{36}$$

$$e_2 = \frac{1}{1 + \phi \bar{\operatorname{tr}} \Sigma_2 (\lambda I_d + p_1 e_1 \Sigma_1 + p_2 e_2 \Sigma_2)^{-1}}. \tag{37}$$

---

[1]All the linear pencils in this work are very big and are omitted for brevity.

Putting things together gives

$$r^{(1)} \simeq d \cdot G_{1,5} = p_1 e_1 \operatorname{tr} A\Sigma_1 (p_1 e_1 \Sigma_1 + p_2 e_2 \Sigma_2 + \lambda I_d)^{-1} = p_1 \operatorname{tr} A\Sigma_1 K^{-1}.$$

In particular, in the limit $p_2 \to 0^+$ (i.e single data source), the first equation becomes

$$1 - \lambda/\kappa_1 = 1 - \eta_1 \lambda = \phi_1 \eta_1 \bar{\operatorname{tr}}\, \Sigma_1 (I_d + p_1 \eta_1 \Sigma_1)^{-1}$$
$$= \phi_1 \bar{\operatorname{tr}}\, \Sigma_1 (\kappa_1 I_d + \Sigma_1)^{-1},$$

or equivalently,

$$\kappa_1 - \lambda \simeq \kappa_1 \frac{\mathrm{df}_1(\kappa_1; \Sigma_1)}{n_1}. \tag{38}$$

Furthermore, $r^{(1)}$ is now given by

$$r^{(1)} \simeq e_1 \operatorname{tr} A\Sigma_1 (e_1 \Sigma_1 + \lambda I_d)^{-1} = \operatorname{tr} A\Sigma_1 (\Sigma_1 + \kappa_1 I_d)^{-1}. \tag{39}$$

**Computing $r^{(4)}$.** Here, the minimal linear pencil for the random matrix involved $R = AM_1(M + \lambda I_d)^{-1} B (M + \lambda I_d)^{-1}$ is a $16 \times 16$ block matrix $Q$ such that $R = Q^{-1}[1,9]/\lambda$. Thus, $r^{(4)}/d \simeq G_{1,16}/\lambda$, where $G = (\mathrm{id} \otimes \mathbb{E}\,\bar{\operatorname{tr}})[Q^{-1}] \in M_{16}(\mathbb{C})$.

First consider the special case $p_2 \to 0^+$ (i.e $n_2$ is negligible compared to $n_1$). The fixed-point equations defining $G_{1,9}$ are given by

$$G_{1,9} = \lambda \bar{\operatorname{tr}}\, A\Sigma_1 (G_{3,3} B + G_{3,11} I_d)(\lambda I_d + G_{3,3}\Sigma_1)^{-1}(\lambda I_d + G_{11,11}\Sigma_1)^{-1}, \tag{40}$$

$$G_{3,3} = \frac{\lambda}{\lambda - \phi G_{4,2}}, \tag{41}$$

$$G_{11,11} = \frac{\lambda}{\lambda - \phi G_{12,10}}, \tag{42}$$

$$G_{3,11} = \frac{\lambda \phi G_{4,10}}{(\lambda - \phi G_{4,2})(\lambda - \phi G_{12,10})} = \frac{\phi G_{3,3} G_{11,11} G_{4,10}}{\lambda}, \tag{43}$$

$$G_{12,10} = -\lambda \bar{\operatorname{tr}}\, \Sigma_1 (\lambda I_d + G_{11,11}\Sigma_1)^{-1}, \tag{44}$$

$$G_{4,10} = -\lambda \bar{\operatorname{tr}}\, \Sigma_1 (\lambda B - G_{3,11}\Sigma_1)(\lambda I_d + G_{3,3}\Sigma_1)^{-1}(\lambda I_d + G_{11,11}\Sigma_1)^{-1}, \tag{45}$$

$$G_{4,2} = -\lambda \bar{\operatorname{tr}}\, \Sigma_1 (\lambda I_d + G_{3,3}\Sigma_1)^{-1}. \tag{46}$$

Observe the equations for $G_{3,11}$ and $G_{4,10}$ further give $G_{3,11} = -v$, where $v$ solves the equation

$$v = \phi G_{3,3} G_{11,11} \bar{\operatorname{tr}}\, \Sigma_1 (v\Sigma_1 + \lambda B)(\lambda I_d + G_{3,3}\Sigma_1)^{-1}(\lambda I_d + G_{11,11}\Sigma_1)^{-1}. \tag{47}$$

Now, let $e$ be the unique non-negative solution to the equation

$$e = \frac{1}{1 + \phi \bar{\operatorname{tr}}\, \Sigma_1 (\lambda I_d + e\Sigma_1)^{-1}}. \tag{48}$$

It is easy to see that we must have $G_{3,3} = G_{11,11} = e$ and

$$r^{(4)}/d = \frac{G_{1,9}}{\lambda} = \bar{\operatorname{tr}}\, A\Sigma_1 (eB - vI_d)(\lambda I_d + \Sigma_1)^{-2}$$
$$= e^{-1}\bar{\operatorname{tr}}\, AB\Sigma_1 (\kappa I_d + \Sigma_1)^{-2} - ve^{-2}\bar{\operatorname{tr}}\, A\Sigma_1 (\kappa I_d + \Sigma_1)^{-2} \tag{49}$$
$$= \frac{\kappa}{\lambda}\bar{\operatorname{tr}}\, AB\Sigma_1 (\kappa I_d + \Sigma_1)^{-2} - \frac{v\kappa^2}{\lambda^2}\bar{\operatorname{tr}}\, A\Sigma_1 (\kappa I_d + \Sigma_1)^{-2},$$

where $\kappa := \lambda/e$. Furthermore, $v$ defined earlier now satisfies

$$v = \phi e^2 \bar{\operatorname{tr}}\, \Sigma_1 (v\Sigma_1 + \lambda B)(\lambda I_d + e\Sigma_1)^{-2}$$
$$= \phi \bar{\operatorname{tr}}\, \Sigma_1 (v\Sigma_1 + \lambda B)(\kappa I_d + \Sigma_1)^{-1}.$$

Solving for $v$ gives

$$v = \frac{\phi \lambda \bar{\operatorname{tr}}\, B\Sigma_1 (\kappa I_d + \Sigma_1)^{-2}}{1 - \phi \bar{\operatorname{tr}}\, \Sigma_1^2 (\kappa I_d + e\Sigma_1)^{-2}} \simeq \frac{\lambda \operatorname{tr} B\Sigma_1 (\kappa I_d + \Sigma_1)^{-2}}{n - \mathrm{df}_2(\kappa)}.$$

In particular, if $B = \Sigma_1$ and $A = I_d$, then

$$v = \frac{\lambda \, df_2(\kappa)}{n - df_2(\kappa)},$$

and so we must have

$$
\begin{aligned}
r^{(4)}/d = \frac{G_{1,9}}{\lambda} &= \frac{\kappa}{\lambda} \bar{tr} \, \Sigma_1^2 (\kappa I_d + \Sigma_1)^{-2} - \frac{v\kappa^2}{\lambda^2} \bar{tr} \, \Sigma_1 (\kappa I_d + \Sigma_1)^{-2} \\
&= \frac{\kappa}{\lambda} \frac{1}{d} \, df_2(\kappa) - \frac{\kappa^2}{\lambda} \frac{1}{d} \, tr \, \Sigma_1 (\kappa I_d + \Sigma_1)^{-2} \cdot \frac{df_2(\kappa)}{n - df_2(\kappa)} \\
&= \frac{\kappa}{\lambda} \frac{1}{d} \, df_2(\kappa) - \frac{\kappa}{\lambda} \frac{1}{d} \cdot (df_1(\kappa) - df_2(\kappa)) \cdot \frac{df_2(\kappa)}{n - df_2(\kappa)} \\
&= \frac{\kappa}{\lambda} \frac{1}{d} (n - df_1(\kappa)) \cdot \frac{df_2(\kappa)}{n - df_2(\kappa)} \\
&\simeq \frac{n}{d} \frac{df_2(\kappa)}{n - df_2(\kappa)} \simeq \frac{1}{\phi} \frac{df_2(\kappa)}{n - df_2(\kappa)},
\end{aligned}
\tag{50}
$$

where, in the last 2 steps we have made use of the following identities which follow from the definition of $\kappa$

$$\kappa - \lambda \simeq \frac{\kappa \, df_1(\kappa)}{n},$$

$$\kappa \, tr \, \Sigma_1 (\kappa I_d + \Sigma_1)^{-2} = df_1(\kappa) - df_2(\kappa).$$

We deduce that the variance term in the bias-variance decomposition of the test error is given by

$$Var = \sigma^2 \frac{1}{n} r^{(4)} \simeq \sigma^2 \frac{df_2(\kappa)}{n - df_2(\kappa)} = \sigma^2 u = \sigma^2 \frac{df_2(\kappa)/n}{1 - df_2(\kappa)/n}. \tag{51}$$

Let us now compute the limiting value of $r^{(4)}$ for any values of the proportions $p_1, p_2 \in (0, 1)$ with $p_1 + p_2 = 1$. The fixed-point equations defining $G_{1,9}$ now become

$$G_{1,9} = p_1 \bar{tr} \, A\Sigma_1 S (\lambda I_d + p_1 G_{2,2} \Sigma_1 + p_2 G_{6,6} \Sigma_2)^{-2}, \tag{52}$$

$$\text{with } S := \lambda (G_{2,2} B + G_{2,10} I_d) + p_2 (e_2 G_{2,10} - e_1 G_{6,13}) \Sigma_2, \tag{53}$$

$$G_{2,2} = e_1, \tag{54}$$

$$G_{6,6} = e_2, \tag{55}$$

$$G_{2,10} = G_{3,11} = \frac{\phi e_1^2 G_{4,10}}{\lambda}, \tag{56}$$

$$G_{6,13} = G_{7,14} = \frac{\phi e_2^2 G_{8,13}}{\lambda}, \tag{57}$$

$$G_{8,13} = \lambda \bar{tr} \, \Sigma_2 (\lambda B - p_1 G_{3,11} \Sigma_1 - p_2 G_{7,14} \Sigma_2)(\lambda I_d + p_1 G_{3,3} \Sigma_1 + p_2 G_{7,7} \Sigma_2)^{-2}, \tag{58}$$

$$G_{4,10} = \lambda \bar{tr} \, \Sigma_1 (\lambda B - p_1 G_{3,11} \Sigma_1 - p_2 G_{7,14} \Sigma_2)(\lambda I_d + p_1 G_{3,3} \Sigma_1 + p_2 G_{7,7} \Sigma_2)^{-2}, \tag{59}$$

where $e_1 \geq 0$ and $e_2 \geq 0$ solve the following system of equations

$$e_1 = \frac{1}{1 + \phi \bar{tr} \, \Sigma_1 (\lambda I_d + p_1 e_1 \Sigma_1 + p_2 e_2 \Sigma_2)^{-2}}, \tag{60}$$

$$e_2 = \frac{1}{1 + \phi \bar{tr} \, \Sigma_2 (\lambda I_d + p_1 e_1 \Sigma_1 + p_2 e_2 \Sigma_2)^{-2}}. \tag{61}$$

Furthermore, we deduce that $G_{6,13} = -v_2$ and $G_{2,10} = -v_1$, where $v_1$ and $v_2$ solve the equations

$$v_1 = \phi e_1^2 \bar{tr} \, \Sigma_1 (p_1 v_1 \Sigma_1 + p_2 v_2 \Sigma_2 + \lambda B)(\lambda I_d + p_1 e_1 \Sigma_1 + p_2 e_2 \Sigma_2)^{-2}, \tag{62}$$

$$v_2 = \phi e_2^2 \bar{tr} \, \Sigma_2 (p_1 v_1 \Sigma_1 + p_2 v_2 \Sigma_2 + \lambda B)(\lambda I_d + p_1 e_1 \Sigma_1 + p_2 e_2 \Sigma_2)^{-2}. \tag{63}$$

Putting things together gives the formula for $r^{(4)}$ proposed in Proposition 1. $\qquad\square$

In particular, taking $p_2 \to 0$ (i.e $p_1 \to 1$) recovers the formula as a special case.

**Computing $r^{(3)}$.** A minimal linear pencil for the corresponding random matrix $R = AM_1(M + \lambda I_d)^{-1}B(M + \lambda I_d)^{-1}M_1$ is a $17 \times 17$ block matrix $Q$ such that $R = Q^{-1}[1, 16]$. This gives

$$r^{(3)}/d \simeq G_{1,16},$$

where $G = (\mathrm{id} \otimes \mathbb{E}\,\bar{\mathrm{tr}}\,)[Q^{-1}] \in M_{17}(\mathbb{C})$. The fixed-point eqautions that determine $G_{1,16}$ are

$$G_{1,16} = p_1 \bar{\mathrm{tr}}\, A\Sigma_1 S(\lambda I_d + p_1 e_1 \Sigma_1 + p_2 e_2 \Sigma_2)^{-2}$$
$$\text{with } S := p_1 e_1^2 (\lambda B - p_2 G_{6,13}\Sigma_2)\Sigma_1 - G_{2,10}(\lambda I_d + p_2 e_2 \Sigma_2)^2,$$
$$G_{7,14} = G_{6,13} = -v_2,$$
$$G_{3,11} = G_{2,10} = -v_1.$$

We deduce the formula given in Proposition 1. In particular, taking the limit $p_2 \to 0$ (i.e $p_1 \to 1$) gives

- $\widetilde{S} \simeq e_1^2 B\Sigma_1 + \lambda v_1 I_d = e_1^2 B\Sigma_1 + \lambda^2 u_1 I_d,$
- $v_1 = \phi e_1^2 \bar{\mathrm{tr}}\, \Sigma_1(v_1\Sigma_1 + \lambda B)(e_1\Sigma_1 + \lambda I_d)^{-2} = \phi\bar{\mathrm{tr}}\, \Sigma(v_1\Sigma_1 + \lambda B)(\Sigma + \kappa_1 I_d)^{-2},$ i.e

$$u_1 = \frac{v_1}{\lambda} \quad = \frac{\phi\bar{\mathrm{tr}}\, B\Sigma_1(\Sigma_1 + \kappa I_d)^{-2}}{1 - \phi\bar{\mathrm{tr}}\, \Sigma_1^2(\Sigma_1 + \kappa_1 I_d)^{-2}} \simeq \frac{\mathrm{tr}\, B\Sigma_1(\Sigma_1 + \kappa I_d)^{-2}}{n - \mathrm{df}_2^{(1)}(\kappa_1)}. \tag{64}$$

Finally, recalling that $\kappa_1 = \lambda/e_1$ by construction, we get

$$r^{(3)} \simeq d \cdot G_{1,16} = e_1^2 \,\mathrm{tr}\, AB\Sigma_1^2(e_1\Sigma_1 + \lambda I_d)^{-2} + \lambda^2 u_1 \bar{\mathrm{tr}}\, A\Sigma_1(e_1\Sigma_1 + \lambda I_d)^{-2}$$

$$= \mathrm{tr}\, AB\Sigma_1^2(\Sigma_1 + \kappa_1 I_d)^{-2} + \frac{\lambda^2 u_1}{e_1^2}\,\mathrm{tr}\, A\Sigma_1(\Sigma_1 + \kappa_1 I_d)^{-2}$$

$$\simeq \mathrm{tr}\, AB\Sigma_1^2(\Sigma_1 + \kappa_1 I_d)^{-2} + \kappa_1^2 \,\mathrm{tr}\, A\Sigma_1(\Sigma_1 + \kappa_1 I_d)^{-2} \cdot \frac{\mathrm{tr}\, B\Sigma_1(\Sigma_1 + \kappa I_d)^{-2}}{n - \mathrm{df}_2^{(1)}(\kappa_1)}.$$

**Computing $r^{(2)}$.** A pencil for the relevant matrix $R = \lambda^2 A(M + \lambda I_d)^{-1}B(M + \lambda I_d)^{-1}$ has minimal linear pencil $Q$ of size $15 \times 15$, where $R = Q^{-1}[1, 8]$. We deduce that $r^{(2)}/d = \mathbb{E}\,\bar{\mathrm{tr}}\, R/\lambda^2 = G_{1,8}/\lambda^2$, where $G = (\mathrm{id} \otimes \mathbb{E}\,\bar{\mathrm{tr}}\,)Q^{-1} \in M_{15}(\mathbb{C})$. The fixed-point equations defining $G_{1,8}$ are given by

$$G_{1,8} = \lambda\bar{\mathrm{tr}}\, AS(p_1 G_{2,2}\Sigma_1 + p_2 G_{5,5}\Sigma_2 + \lambda I_d)^{-2}, \tag{65}$$
$$\text{with } S = \lambda B - p_1 G_{2,9}\Sigma_1 - p_2 G_{5,12}\Sigma_2, \tag{66}$$
$$G_{2,2} = e_1, \tag{67}$$
$$G_{5,5} = e_2, \tag{68}$$
$$G_{2,9} = G_{3,10} = \frac{\phi e_1^2 G_{4,9}}{\lambda}, \tag{69}$$
$$G_{5,12} = G_{6,13} = \frac{\phi e_2^2 G_{7,12}}{\lambda}, \tag{70}$$
$$G_{4,9} = -\lambda\bar{\mathrm{tr}}\, \Sigma_1(\lambda B - p_1 G_{3,10}\Sigma_1 - p_2 G_{6,13}\Sigma_2)(p_1 G_{2,2}\Sigma_1 + p_2 G_{5,5}\Sigma_2 + \lambda I_d)^{-2}, \tag{71}$$
$$G_{7,12} = -\lambda\bar{\mathrm{tr}}\, \Sigma_2(\lambda B - p_1 G_{3,10}\Sigma_1 - p_2 G_{6,13}\Sigma_2)(p_1 G_{2,2}\Sigma_1 + p_2 G_{5,5}\Sigma_2 + \lambda I_d)^{-2}. \tag{72}$$

Based on previous calculations, we deduce that $G_{2,9} = -v_1$ and $G_{5,12} = -v_2$, and so

$$r^{(2)} \simeq d \cdot \frac{G_{1,8}}{\lambda^2} = \frac{1}{\lambda}\,\mathrm{tr}\, A(p_1 v_1 \Sigma_1 + p_2 v_2 \Sigma_2 + \lambda B)(p_1 e_1 \Sigma_1 + p_2 e_2 \Sigma_2 + \lambda I_d)^{-2} = \mathrm{tr}\, A\widetilde{L}K^{-2},$$

as claimed. This completes the proof of Proposition 1. $\qquad\square$

### E.4  PROOF OF PROPOSITION 2

In Section G.1 we will establish a more general result which implies Proposition 2 as a special case.

## F    PROOF OF THEOREM 1 AND COROLLARIES

Let us note that the results in Bach (2023) were obtained in a two-stage approach, where random matrix theory is applied on the raw (unquenched test error $\|\widehat{w} - w_1\|_{\Sigma}^2$ with the projection matrix treated like a deterministic matrix, and then RMT is done one more on the resulting expressions but now treating $S$ as random. The case general case $p_2 \in (0,1)$ is much more difficult; the key technical difficulty can be pinned down to the problem of obtaining analytic deterministic equivalents for the trace of the and derivatives of the resolvent of a sum of random matrices. To circumvent this, we employ the tools of operator-valued free probability theory.

### F.1    PROOF OF THEOREM 1

From Proposition 4 and 1 applied with $\Sigma_1 = \Sigma_2 = \Sigma$, we know that

$$E_{test}(\widehat{f}_{CL}) = V_1 + B_1, \text{ with}$$

$$
\begin{aligned}
V_1 &\simeq \sum_{j=1}^{2} p_j \sigma_j^2 \frac{1}{n} \operatorname{tr} \Sigma_k D_{j,k} K^{-2} = \sum_{j=1}^{2} p_j \sigma_j^2 \frac{\kappa}{\lambda} \cdot \frac{1}{n} \operatorname{tr} \Sigma(\Sigma - \kappa u I_d)(\Sigma + \kappa I_d)^{-2} \\
&= \sigma^2 \frac{\kappa}{\lambda} \cdot \frac{1}{n} \operatorname{tr} \Sigma(\Sigma - \kappa u I_d)(\Sigma + \kappa I_d)^{-2}, \\
B_1 &= p_2 \operatorname{tr} \Delta \Sigma_2 C_{2,1} K^{-2} + \lambda^2 \operatorname{tr} \Gamma L_1' K^{-2} \\
&= p_2 \operatorname{tr} \Delta \Sigma \left( p_2(1 + p_1 u)\Sigma^2 + u(p_1 \Sigma + \kappa I_d)^2 \right) (\Sigma + \kappa I_d)^{-2} + \kappa^2 (u+1) \operatorname{tr} \Gamma \Sigma(\Sigma + \kappa I_d)^{-2}.
\end{aligned}
$$

Now, for the $V_1$ term, first observe that

$$
\begin{aligned}
\operatorname{tr} \Sigma(\Sigma - \kappa u I_d)(\Sigma + \kappa I_d)^{-2} &= \operatorname{tr} \Sigma\left(\Sigma - \frac{\kappa \, \mathrm{df}_2}{n - \mathrm{df}_2} I_d\right)(\Sigma + \kappa I_d)^{-2} \\
&= \mathrm{df}_2 - \frac{\mathrm{df}_2}{n - \mathrm{df}_2} \cdot \kappa \operatorname{tr} \Sigma(\Sigma + \kappa I_d)^{-2} \\
&= \mathrm{df}_2 - \frac{\mathrm{df}_2}{n - \mathrm{df}_2}(\mathrm{df}_1 - \mathrm{df}_2) \\
&= \frac{\mathrm{df}_2}{n - \mathrm{df}_2}(n - \mathrm{df}_1).
\end{aligned}
$$

We deduce that

$$V_1 = \sigma^2 \cdot (1 - \mathrm{df}_1 /n)\frac{\kappa}{\lambda} \cdot \frac{\mathrm{df}_2}{n - \mathrm{df}_2} = \sigma^2 \cdot \frac{\mathrm{df}_2}{n - \mathrm{df}_2} =: V,$$

where we have used the identity $\kappa - \lambda = \kappa \, \mathrm{df}_1 /n$, which defines $\kappa$.

We now handle the $B_1$ term. First observe that $u + 1 = n/(n - \mathrm{df}_2)$, and so one computes

$$\kappa^2 (u+1) \operatorname{tr} \Gamma \Sigma(\Sigma + \kappa I_d)^{-2} = \kappa^2 \frac{n}{n - \mathrm{df}_2} \operatorname{tr} \Gamma \Sigma(\Sigma + \lambda I_d)^{-2} =: B,$$

which is the classical formula for the bias term.

To finalize, observe that

$$
\begin{aligned}
\operatorname{tr} \Delta \Sigma C_{2,1} K^{-2} &= \operatorname{tr} \Delta \Sigma \left( p_2(1 + p_1 u)\Sigma^2 + u(p_1 \Sigma + \kappa I_d)^2 \right)(\Sigma + \kappa I_d)^{-2} \\
&= p_2(1 + p_1 u) \operatorname{tr} \Delta \Sigma^3 (\Sigma + \kappa I_d)^{-2} + u \operatorname{tr} \Delta \Sigma(p_1 \Sigma + \kappa I_d)^2(\Sigma + \kappa I_d)^{-2} =: \zeta,
\end{aligned}
$$

which concludes the proof.    □

### F.2 Proof of Corollary 1

Indeed, here we have $\kappa \to 0$ and $u \to \phi/(1 - \phi)$ in the limit $\lambda \to 0^+$. Theorem 1 then gives

$$E_{test}(\widehat{f}_{CL}) \simeq V + B + \zeta, \text{ where } V = \frac{\sigma^2 \phi}{1 - \phi}, \ B = 0,$$

$$\zeta = \frac{p_2 c^2}{1 - \phi} \left( p_2(1 - \phi + p_1 \phi) + p_1^2 \phi \right) = \frac{p_2 c^2}{1 - \phi} (p_2(1 - p_2 \phi) + p_1^2 \phi)$$

$$= \frac{p_2 c^2}{1 - \phi} (p_2 + (p_1 - p_2)\phi) = p_2^2 c^2 + \frac{p_2 p_1 c^2 \phi}{1 - \phi}.$$

For small $\phi$, this further gives $E_{test}(\widehat{f}_{CL}) \simeq \sigma^2 \phi/(1 - \phi) + p_2^2 c^2 + O(\phi^2) \simeq \sigma^2 d/n + p_2^2 c^2 + O(\phi^2)$. □

### F.3 Proof of Corollary 3

The setup can be seen as a special instance of the setup considered in the proof of Proposition 1 (cf. Appendix F.1), since it corresponds to taking $\Sigma_1 = (1 - \alpha)\Sigma/p_1$, and $\Sigma_2 = \alpha\Sigma/p_2$. We must have

$$\frac{1}{e_1} = 1 + \phi \bar{\text{tr}} \, \Sigma_1 K^{-1} = 1 + \frac{(1 - \alpha)\phi/p_1}{(1 - \alpha)e_1 + \alpha e_2 + \lambda}, \tag{73}$$

$$\frac{1}{e_2} = 1 + \phi \bar{\text{tr}} \, \Sigma_2 K^{-1} = 1 + \frac{\alpha\phi/p_2}{(1 - \alpha)e_1 + \alpha e_2 + \lambda}. \tag{74}$$

At least for $\lambda = 0$ and $0 < \phi < 1$, these equations can be solved explicitly to get $e_1, e_1 \geq 0$ but the resulting formulae are rather complicated, and therefore are omitted altogether. In any case, heorem 1 correctly predicts the test error, as can be seen in Figure 12.

A particular case where things are solvable to give simple expressions, is when $\phi \to 0^+$. In this limit, it is easy to see that $e_1 = e_2 = 1$ and $u_1 = u_2 = 0$. This gives

$$K = \Sigma + \lambda I_d, \tag{75}$$
$$L' = \Sigma, \tag{76}$$
$$C_1 = (1 - \alpha)\Sigma, \tag{77}$$
$$C_2 = \alpha\Sigma, \tag{78}$$
$$D_k = \Sigma, \tag{79}$$
$$\lambda^2 r^{(2)}(A, \Sigma) \simeq \lambda^2 \operatorname{tr} A\Sigma(\Sigma + \lambda I_d)^{-2} = \lambda \cdot \left( \operatorname{tr} A\Sigma(\Sigma + \lambda)^{-1} - \operatorname{tr} A\Sigma^2(\Sigma + \lambda I_d)^{-2} \right), \tag{80}$$
$$r_1^{(3)}(A, \Sigma) \simeq p_1 \operatorname{tr} A\Sigma_1 C_1 K^{-2} = p_1^1 \alpha^2 \operatorname{tr} A\Sigma^2(\Sigma + \lambda I_d)^{-2}, \tag{81}$$
$$r_2^{(3)}(A, \Sigma) \simeq p_2 \operatorname{tr} A\Sigma_2 C_2 K^{-2} = p_2^2(1 - \alpha)^2 \operatorname{tr} A\Sigma^2(\Sigma + \lambda I_d)^{-2}, \tag{82}$$
$$r_1^{(4)}(A, \Sigma) \simeq p_1 \operatorname{tr} A\Sigma_1 D_1 K^{-2} = p_1 \alpha \operatorname{tr} A\Sigma^2(\Sigma + \lambda I_d)^{-2}, \tag{83}$$
$$r_2^{(4)}(A, \Sigma) \simeq p_2 \operatorname{tr} A\Sigma_2 D_2 K^{-2} = p_2(1 - \alpha) \operatorname{tr} A\Sigma^2(\Sigma + \lambda I_d)^{-2}. \tag{84}$$

We deduce that

$$V_1 = \sum_{j=1}^{2} p_j \frac{\sigma_j^2}{n} r_j^{(4)}(I_d, \Sigma) = \left( (1 - \alpha)p_1\sigma_1^2 + \alpha p_2 \sigma_2^2 \right) \frac{\mathrm{df}_2(\lambda; \Sigma)}{n}, \tag{85}$$

$$B_1 = r_2^{(3)}(\Delta, \Sigma) + \lambda^2 r^{(2)}(\Gamma, \Sigma) \xrightarrow{\lambda \to 0^+} p_2^2(1 - \alpha)^2 \operatorname{tr} \Delta. \tag{86}$$

Putting things together then gives

$$E_{test}(\widehat{f}_{CL}) \simeq B_1 + V_1 \simeq p_2^2(1 - \alpha)^2 \operatorname{tr} \Delta + (\alpha p_1 \sigma_1^2 + (1 - \alpha)p_2 \sigma_2^2)\frac{d}{n},$$

as claimed. □

### F.4 PROOF OF COROLLARY 2

Applying the first part of Corollary 1 recursively gives for any iteration $t \geq 1$,

$$E_{test}(\widehat{f}_{CL}^{(t)}) \simeq c_t^2 \simeq \overline{E} + p_2^2 c_{t-1}^2 \simeq \ldots \simeq p_2^{2t} c_0^2 + \frac{1 - p_2^{2t}}{1 - p_2^2} \overline{E}, \text{ with } \overline{E} := \frac{\sigma^2 \phi}{1 - \phi}.$$

Iterating the above gives

$$c_{t+1}^2 \simeq \frac{\sigma^2 \phi_t}{1 - \phi_t} + p_2^2 c_t^2, \quad \phi_t = d/N_t, \quad N_t = n, \quad c_0^2 = c^2. \tag{87}$$

Setting $\overline{E} := \sigma^2 \phi/(1 - \phi) \simeq \sigma^2 d/n$, we get

$$
\begin{aligned}
E_{test}(\widehat{f}_{CL}^{(t+1)}) \simeq c_{t+1}^2 &\simeq p_2^2 c_t^2 + \frac{\sigma^2 \phi_t}{1 - \phi_t} \\
&\simeq p_2^2 \left( p_2^2 c_{t-1}^2 + \frac{\sigma^2 \phi_{t-1}}{1 - \phi_{t-1}} \right) + \frac{\sigma^2 \phi_t}{1 - \phi_t} \\
&\simeq p_2^{2(1+1)} c_{t-1}^2 + p_2^2 \frac{\sigma^2 \phi_{t-1}}{1 - \phi_{t-1}} + \frac{\sigma^2 \phi_t}{1 - \phi_t} \\
&\vdots \\
&\simeq p_2^{2(t+1)} c_0^2 + \sum_{0 \leq j \leq t} \frac{\sigma^2 \phi_j}{1 - \phi_j} p_2^{2(t-j)} \\
&= p_2^{2(t+1)} c^2 + \overline{E} \sum_{0 \leq j \leq t} p_2^{2j} \\
&= p_2^{2(t+1)} c^2 + \frac{1 - p_2^{2(t+1)}}{1 - p_2^2} \overline{E}.
\end{aligned}
$$

In particular, we if $p_2$ is bounded away from 1 (i.e if $p_1 := 1 - p_2 = \Omega(1)$), we get

$$E_{test}(\widehat{f}_{CL}^{(t)}) \simeq \frac{1}{1 - p_2^2} \overline{E} + p_2^{2t} c^2,$$

for large $t$. The first part is just a constant multiple of the scaling law we would have with training on a dataset comprising of $n$ units of clean data.

On the other hand, we have

$$\lim_{p_2 \to 1} E_{test}(\widehat{f}_{CL}^{(t)}) \simeq c^2 + t\overline{E}.$$

This is an increasing function of $t$, lower-bounded by $c^2 + \overline{E}$. We recover the classical picture, in which model collapse prevails (depending on the size of $c^2$, as per Corollary 1). □

### F.5 PROOF OF COROLLARY 4

From Theorem 1 and the observation that $\kappa \to \phi - 1$ and $u \to 1/(1 - \phi/\phi^2) = \phi/(\phi - 1)$ in the limit $\lambda \to 0^+$, we have $E_{test}(\widehat{w}) \simeq \overline{E} + \zeta$, with

$$\overline{E} = V + B, \quad B = r^2 \frac{(\phi - 1)^2}{\phi^2} \frac{1}{1 - 1/\phi} = r^2 (1 - 1/\phi), \quad V = \frac{\sigma^2}{\phi - 1},$$

$$\zeta = \frac{p_2 c^2}{\phi^2} \left( p_2(1 + \frac{p_1}{\phi - 1}) + (p_1 + \phi - 1)^2 \right),$$

and the first part of the result follows after some algebra.

The second part then follows from expanding the above around $\phi = \infty$. □

# G   PROOF OF PROPOSITION 2 AND THEOREM 2

## G.1   PROOF OF PROPOSITION 2

We state and proof a more general result without the requirement $\Sigma_1 = \Sigma_2 = \Sigma$.

Let $(e_1, e_2, \tau)$ be the unique nonnegative solution to the following fixed-point equations

$$e_1 = \frac{1}{1 + \psi\tau\bar{\text{tr}}\,\Sigma_1 K^{-1}}, \tag{88}$$

$$e_2 = \frac{1}{1 + \psi\tau\bar{\text{tr}}\,\Sigma_2 K^{-1}}, \tag{89}$$

$$\tau = \frac{1}{1 + \bar{\text{tr}}\,K_0 K^{-1}}, \tag{90}$$

$$\text{with } K_0 := p_1 e_1 \Sigma_1 + p_2 e_2 \Sigma_2, \ K := \gamma\tau K_0 + \lambda I_d. \tag{91}$$

Also, let $(v_1, v_2, \omega)$ to be the unique nonnegative solution to the following fixed-point equations

$$v_1 = \psi e_1^2 \bar{\text{tr}}\,\Sigma_1 (\gamma\tau^2 L + \lambda\omega I_d) K^{-2}, \tag{92}$$

$$v_2 = \psi e_2^2 \bar{\text{tr}}\,\Sigma_2 (\gamma\tau^2 L + \lambda\omega I_d) K^{-2}, \tag{93}$$

$$\omega = \tau^2 \bar{\text{tr}}\,(\gamma K_0^2 + \lambda L) K^{-2}, \tag{94}$$

$$\text{with } L := p_1 v_1 \Sigma_1 + p_2 v_2 \Sigma_2 + \lambda B. \tag{95}$$

Finally, define $d \times d$ matrices $C_1, C_2, D_1, D_2, E$ by

$$C_1 := \gamma p_1 e_1^2 \big(\gamma\tau^2(B + p_2 u_2 \Sigma_2) + \omega I_d\big)\Sigma_1 + u_1(\gamma\tau p_2 e_2 \Sigma_2 + \lambda I_d)^2, \tag{96}$$

$$C_2 := \gamma p_2 e_2^2 \big(\gamma\tau^2(B + p_1 u_1 \Sigma_1) + \omega I_d\big)\Sigma_2 + u_2(\gamma\tau p_1 e_1 \Sigma_1 + \lambda I_d)^2, \tag{97}$$

$$D_1 := \tau^2 e_1 B + (e_1\omega - \tau v_1)I_d + \gamma\tau^2 p_2(e_1 u_2 - e_2 u_1)\Sigma_2, \tag{98}$$

$$D_2 := \tau^2 e_2 B + (e_2\omega - \tau v_2)I_d + \gamma\tau^2 p_1(e_2 u_1 - e_1 u_2)\Sigma_1, \tag{99}$$

$$E := \gamma(\gamma\tau^2 B + \omega I_d), \tag{100}$$

**Proposition 3.** *In the proportionate scaling limit* (6)*, it holds that*

$$r_j^{(1)}(A) \simeq \gamma\tau p_j e_j \,\text{tr}\, A\Sigma_j K^{-1}, \tag{101}$$

$$r_j^{(3)}(A, B) \simeq \gamma p_j A\Sigma_j C_j K^{-2}, \tag{102}$$

$$r_j^{(4)}(A, B) \simeq \gamma p_j \,\text{tr}\, A\Sigma_j D_j K^{-2}, \tag{103}$$

$$r^{(5)}(A, B) \simeq \text{tr}\, AEK^{-2}. \tag{104}$$

Observe that if we force $\tau = \gamma = 1$ and $\omega = 0$, then we recover the corresponding formulae given in Proposition 1. On the other hand, taking $\Sigma_1 = \Sigma_2 = \Sigma$ gives Proposition 2.

*Proof.* WLOG, we only consider the cases where $j = 1$.

**Computing $r_1^{(1)}$.** There is a $11 \times 11$ minimal linear pencil $Q$ such that $ASRS^\top M_1 = Q^{-1}[1, 10]$ (zero-based indexing). We deduce that

$$r_1^{(1)} := \mathbb{E} \,\text{tr}\, ASRS^\top M_1 \simeq d \cdot G_{1,10}, \tag{105}$$

where $G := (\text{id} \otimes \mathbb{E}\,\bar{\text{tr}})Q^{-1} \in \mathbb{C}^{11 \times 11}$. Moreover, $G_{1,10}$ is given by the following fixed-point equations

$$G_{1,10} = p_1 \gamma G_{2,2} G_{5,5} \bar{\text{tr}}\, A\Sigma_1 K^{-1}, \tag{106}$$

$$\text{with } K := \gamma G_{2,2} L + \lambda I_d, \ L := p_1 G_{5,5}\Sigma_1 + p_2 G_{8,8}\Sigma_2, \tag{107}$$

$$G_{5,5} = \frac{1}{1 + \phi\gamma G_{2,2}\bar{\text{tr}}\,\Sigma_1 K^{-1}} = \frac{1}{1 + \psi G_{2,2}\bar{\text{tr}}\,\Sigma_1 K^{-1}}, \tag{108}$$

$$G_{8,8} = \frac{1}{1 + \phi\gamma G_{2,2}\bar{\text{tr}}\,\Sigma_2 K^{-1}} = \frac{1}{1 + \psi G_{2,2}\bar{\text{tr}}\,\Sigma_2 K^{-1}}, \tag{109}$$

$$G_{2,2} = \frac{1}{1 + \bar{\text{tr}}\,LK^{-1}}, \tag{110}$$

Then, one deduces that

$$\operatorname{tr} ASRS^\top M_1 \simeq d \cdot G_{1,10} = p_1 e_1 \tau \gamma \operatorname{tr} A\Sigma_1 K^{-1}. \tag{111}$$

**Computing $r_1^{(4)}$.** Here, the pencil $Q$ is $20 \times 20$ and $AM_1SRS^\top SRS^\top = -Q^{-1}[1,13]/\lambda$. We deduce that

$$r_1^{(4)} := \mathbb{E} \operatorname{tr} AM_1SRS^\top BSRS^\top \simeq -d \cdot G_{1,13}/\lambda, \tag{112}$$

where $G := (\operatorname{id} \otimes \mathbb{E} \operatorname{\bar{t}r})Q^{-1} \in \mathbb{C}^{20\times20}$. Moreover, $G_{1,13}$ is given by the following fixed-point equations

$$-G_{1,13} = p_1\gamma\operatorname{\bar{t}r} A\Sigma_1 TK^{-2}, \text{ where} \tag{113}$$

$$T := \lambda(\tau^2 e_1 B + (e_1 G_{6,12} + \tau G_{3,15})I_d) + p_2\gamma\tau^2(e_2 G_{3,15} - e_1 G_{9,18})\Sigma_2, \tag{114}$$

$$G_{12,12} = G_{6,6} = \tau, \tag{115}$$

$$G_{3,15} = \frac{\phi e_1^2 G_{4,14}}{\lambda}, \tag{116}$$

$$G_{4,14} = -\lambda\gamma\operatorname{\bar{t}r} \Sigma_1 \left(\gamma\tau^2(p_1 G_{3,15}\Sigma_1 + p_2 G_{9,18}\Sigma_2) - \lambda(\gamma\tau^2 B + G_{6,12}I_d)\right) K^{-2}, \tag{117}$$

$$G_{9,18} = \frac{\phi e_2^2 G_{10,17}}{\lambda}, \tag{118}$$

$$G_{10,17} = -\lambda\gamma\operatorname{\bar{t}r} \Sigma_2 \left(\gamma\tau^2(p_1 G_{3,15}\Sigma_1 + p_2 G_{9,18}\Sigma_2) - \lambda(\gamma\tau^2 B + G_{6,12}I_d)\right) K^{-2}, \tag{119}$$

$$G_{6,12} = -\tau^2 G_{7,11}, \tag{120}$$

$$G_{7,11} = -\operatorname{\bar{t}r} \left(\gamma K_0^2 + \lambda(\lambda B - p_1 G_{3,15}\Sigma_1 - p_2 G_{9,18}\Sigma_2)\right)K^{-2}, \tag{121}$$

We deduce that $G_{3,15} = -v_1$, $G_{9,18} = -v_2$, and $G_{6,12} = \omega$, where $v_1, v_2, \omega \geq 0$ solve the following fixed-point equations

$$v_1 = \phi\gamma e_1^2\operatorname{\bar{t}r} \Sigma_1 \left(\gamma\tau^2(p_1 v_1\Sigma_1 + p_2 v_2\Sigma_2) + \lambda(\gamma\tau^2 B + \omega I_d)\right) K^{-2}$$
$$= \psi e_1^2\operatorname{\bar{t}r} \Sigma_1(\gamma\tau^2 L + \lambda\omega I_d)K^{-2},$$
$$v_2 = \phi\gamma e_2^2\operatorname{\bar{t}r} \Sigma_2 \left(\gamma\tau^2(p_1 v_1\Sigma_1 + p_2 v_2\Sigma_2) + \lambda(\gamma\tau^2 B + \omega I_d)\right) K^{-2}$$
$$= \psi e_2^2\operatorname{\bar{t}r} \Sigma_2(\gamma\tau^2 L + \lambda\omega I_d)K^{-2},$$
$$\omega = \tau^2\operatorname{\bar{t}r} \left(\gamma K_0^2 + \lambda(\lambda B + p_1 v_1\Sigma_1 + p_2 v_2\Sigma_2)\right)K^{-2} = \tau^2\operatorname{\bar{t}r} \left(\gamma K_0^2 + \lambda L\right)K^{-2},$$

with $L := p_1 v_1\Sigma_1 + p_1 v_2\Sigma_2 + \lambda B$. Putting everything together then gives

$$r_j^{(4)} \simeq -\frac{d \cdot G_{1,13}}{\lambda} = p_1\gamma \operatorname{tr} A\Sigma_1\widetilde{T}K^{-2}, \text{ where}$$
$$\widetilde{T} := T/\lambda = \tau^2 e_1 B + (e_1\omega - \tau v_1)I_d + p_2\gamma\tau^2(e_1 u_2 - e_2 u_1)\Sigma_2 =: D_1.$$

**Computing $r_1^{(3)}$.** The matrix of interest $AM_1SRS^\top BSRS^\top M_1$ admits a minimal linear pencil $Q$ of size $21 \times 21$, such that the formal equals to $Q^{-1}[1,20]$. It follows that

$$r_1^{(3)} := \mathbb{E} \operatorname{tr} AM_1SRS^\top BSRS^\top M_1 \simeq d \cdot G_{1,20}, \tag{122}$$

where $G := (\operatorname{id} \otimes \mathbb{E} \operatorname{\bar{t}r})Q^{-1} \in \mathbb{C}^{21\times21}$. The fixed-point equations defining $G_{1,20}$ are

$$G_{1,20} = p_1\operatorname{\bar{t}r} A\Sigma_1(T/\lambda)K^{-2}, \text{ where}$$
$$T := p_1 G_{3,3}^2\gamma(\gamma\tau^2(\lambda B - p_2 G_{9,18}\Sigma_2) + \lambda G_{6,12}I_d)\Sigma_1 - G_{3,15}(\gamma\tau p_2 G_{9,9}\Sigma_2 + \lambda I_d)^2,$$
$$G_{3,3} = e_1,$$
$$G_{9,9} = e_2,$$
$$G_{6,12} = \omega,$$
$$G_{3,15} = -v_1,$$
$$G_{9,18} = -v_2.$$

Putting things together gives

$$r_1^{(3)} \simeq d \cdot G_{1,20} = \operatorname{tr} A\Sigma_1\widetilde{T}K^{-2},$$
$$\text{where } \widetilde{T} := T/\lambda = \gamma p_1 e_1^2\left(\gamma\tau^2(B + p_2 u_2\Sigma_2) + \omega I_d\right)\Sigma_1 + u_1(\gamma\tau p_2 e_2\Sigma_2 + \lambda I_d)^2 =: C_1,$$

which completes the proof. $\qquad \square$

### G.2 RECOVERING THEOREM 1 FROM THEOREM 2

Indeed, we have

$$\omega' \to 0, \, \theta \to \kappa, \quad u \to \frac{\phi I_{2,2}(\kappa)}{1 - \phi I_{2,2}(\kappa)} = \frac{\mathrm{df}_2(\kappa)/n}{1 - \mathrm{df}_2(\kappa)/n},$$

for any regularization strength $\lambda > 0$, where $\kappa$ is as defined in equation (8). Refer to Lemma 1. Plugging these limits into the formulae provided in Theorem 2 then recovers Theorem 1.

### G.3 PROOF OF THEOREM 2

This follows directly from Proposition 2 and the computations in Section I.2.

## H PHASE-DIAGRAM FOR RANDOM PROJECTIONS MODEL

### H.1 THE GENERAL REGULARIZED CASE

**Lemma 1.** *The scalars $u$ and $\omega'$ which appear in Theorem 2, and described in Definition 2, solve the following pair of linear equations*

$$\begin{aligned}
u &= \phi I_{2,2}(\theta)(1+u) + \phi I_{1,2}(\theta)\omega', \\
\gamma\omega' &= I_{2,2}(\theta)\omega' + \theta^2 I_{1,2}(\theta)(1+u).
\end{aligned} \tag{123}$$

*Furthermore, the solutions can be explicitly represented as*

$$u = \frac{\phi z}{\gamma - \phi z - I_{2,2}(\theta)}, \quad \omega' = \frac{\theta^2 I_{2,2}(\theta)}{\gamma - \phi z - I_{2,2}(\theta)}, \tag{124}$$

*where $z = I_{2,2}(\theta)(\gamma - I_{2,2}(\theta)) + \theta^2 I_{1,2}(\theta)^2$.*

*In particular, in the limit $\gamma \to \infty$, it holds that*

$$\theta \simeq \kappa, \quad \omega' \to 0, \quad u \simeq \frac{\phi I_{2,2}(\kappa)}{1 - \phi I_{2,2}(\kappa)} \simeq \frac{\mathrm{df}_2(\kappa)/n}{1 - \mathrm{df}_2(\kappa)/n}, \tag{125}$$

*where $\kappa > 0$ is as defined in (8).*

*Proof.* The equations defining these are

$$u = \psi e^2 \bar{\mathrm{tr}}\, \Sigma(\gamma\tau^2 L' + \omega I_d)K^{-2}, \tag{126}$$

$$\omega = \tau^2 \bar{\mathrm{tr}}\, (\gamma\omega K_0^2 + \lambda^2 L')K^{-2}, \tag{127}$$

where $K_0 = e\Sigma$, $K = \gamma\tau K_0 + \lambda I_d$, and $L' := u\Sigma + B$. Further, since $B = \Sigma$, we have $L' = (1+u)\Sigma$. Now, we can rewrite the previous equations like so

$$u = \psi e^2 \bar{\mathrm{tr}}\, \Sigma(\gamma\tau^2(1+u)\Sigma + \omega I_d)K^{-2} = \phi\gamma^2\tau^2 e^2(1+u)\bar{\mathrm{tr}}\, \Sigma^2 K^{-2} + \phi\gamma e^2\omega\bar{\mathrm{tr}}\, \Sigma K^{-2},$$

$$\omega = \tau^2 \bar{\mathrm{tr}}\, (\gamma\omega e^2\Sigma^2 + \lambda^2(1+u)\Sigma)K^{-2} = \gamma\tau^2 e^2\omega\bar{\mathrm{tr}}\, \Sigma^2 K^{-2} + \lambda^2\tau^2(1+u)\bar{\mathrm{tr}}\, \Sigma K^{-2}.$$

This can be equivalently written as

$$u = \phi(1+u)\gamma^2\tau^2 e^2\bar{\mathrm{tr}}\, \Sigma^2 K^{-2} + \phi\omega'\gamma^2\tau^2 e^2\bar{\mathrm{tr}}\, \Sigma K^{-2}, \tag{128}$$

$$\gamma\omega' = \omega'\gamma^2\tau^2 e^2\bar{\mathrm{tr}}\, \Sigma^2 K^{-2} + (1+u)\lambda^2\bar{\mathrm{tr}}\, \Sigma K^{-2}. \tag{129}$$

Now, observe that

$$\tau^2 e^2 \bar{\mathrm{tr}}\, \Sigma^2 K^{-2} = \bar{\mathrm{tr}}\, \Sigma^2(\Sigma + \theta I_d)^{-2}/\gamma^2 = I_{2,2}(\theta)/\gamma^2, \tag{130}$$

$$\tau^2 e^2 \bar{\mathrm{tr}}\, \Sigma K^{-2} = \bar{\mathrm{tr}}\, \Sigma(\Sigma + \theta I_d)^{-2}/\gamma^2 = I_{1,2}(\theta)/\gamma^2, \tag{131}$$

$$\lambda^2 \bar{\mathrm{tr}}\, \Sigma K^{-2} = \theta^2 \bar{\mathrm{tr}}\, \Sigma(\Sigma + \theta I_d)^{-2} = \theta^2 I_{1,2}(\theta), \tag{132}$$

$$e^2 \bar{\mathrm{tr}}\, \Sigma K^{-2} = \bar{\mathrm{tr}}\, \Sigma(\Sigma + \theta I_d)^{-2}/(\gamma\tau)^2 = I_{1,2}(\theta)/(\gamma\tau)^2, \tag{133}$$

$$\tau^2 \bar{\mathrm{tr}}\, \Sigma K^{-2} = \bar{\mathrm{tr}}\, \Sigma(\Sigma + \theta I_d)^{-2}/(\gamma e)^2 = I_{1,2}(\theta)/(\gamma e)^2, \tag{134}$$

where we have used the definition $\theta = \lambda/(\gamma\tau e)$. Thus, $u$ and $\omega$ have limiting values $u$ and $\omega$ respectively, which solve the system of linear equations

$$u = \psi\gamma \cdot \gamma^{-2} I_{2,2}(\theta)(1+u) + \psi\gamma \cdot \gamma^{-2} I_{1,2}\omega' = \phi I_{2,2}(\theta)(1+u) + \phi I_{1,2}(\theta)\omega',$$
$$\gamma\omega' = I_{2,2}(\theta)\omega' + \theta^2 I_{1,2}(\theta)(1+u) = I_{2,2}(\theta)\omega' + \theta^2 I_{1,2}(\theta)(1+u),$$

where we have used the identity $\phi\gamma = \psi$. These correspond exactly to the equations given in the lemma. This proves the first part.

For the second part, indeed, $\tau = 1 - \eta_0/\gamma \to 1$ in the limit $\gamma \to \infty$, and so $\theta \simeq \lambda/(\gamma e)$ which verifies the equation

$$\theta \simeq \lambda + \lambda\psi\bar{\text{tr}}\,\Sigma(\gamma e\Sigma + \lambda)^{-1} = \lambda + \phi \cdot \frac{\lambda}{\gamma e}\bar{\text{tr}}\,\Sigma(\Sigma + \frac{\lambda}{\gamma e}I_d)^{-1} \simeq \lambda + \theta\,\text{tr}\,\Sigma(\Sigma + \theta I_d)^{-1}/n,$$

i.e $\theta \simeq \lambda + \theta\,\text{df}_1(\theta)/n$ and $\theta > 0$. By comparing with the equation $\kappa - \lambda = \kappa\,\text{df}_1(\kappa)/n$ satisfied by $\kappa > 0$ in (8), we conclude $\theta \simeq \kappa$.

Now, the equations (123) become $\omega' = 0$, and $u = \phi I_{2,2}(\kappa)(1+u)$, i.e

$$u = \frac{\phi I_{2,2}(\kappa)}{1 - \phi I_{2,2}(\kappa)} \simeq \frac{\text{df}_2(\kappa)/n}{1 - \text{df}_2(\kappa)/n},$$

as claimed. $\qquad\square$

## H.2 UNREGULARIZED LIMIT

Define the following auxiliary quantities

$$\theta := \frac{\lambda}{\gamma\tau e}, \quad \chi := \frac{\lambda}{\tau}, \quad \kappa := \frac{\lambda}{e}. \tag{135}$$

where $\tau$, $e$, $u$, and $\omega$ are as previously defined in Section 3.2.

**Lemma 2.** *In the limit $\lambda \to 0^+$, we have the following analytic formulae*

$$\chi \to \chi_0 = (1-\psi)_+ \cdot \gamma\theta_0, \tag{136}$$
$$\kappa \to \kappa_0 = (\psi-1)_+ \cdot \theta_0/\phi, \tag{137}$$
$$\tau \to \tau_0 = 1 - \eta_0/\gamma, \tag{138}$$
$$e \to e_0 = 1 - \phi\eta_0. \tag{139}$$

*Proof.* From equations (9) and the constraint $\Sigma_1 = \Sigma_2 = \Sigma$, we know that $e_1 = e_2 = e$, where $e$ and $\tau$ are unique positive solutions to a pair of fixed point equations. Observe that $K_0 = e\Sigma$ and $K = \gamma\tau K_0 + \lambda I_d = \gamma\tau e \cdot (\Sigma + \theta I_d)$. Defining $\eta := I_{1,1}(\theta)$, one can then rewrite the equations defining $e$ and $\tau$ as follows

$$e' = \frac{\lambda}{e} = \lambda + \psi\tau\lambda\bar{\text{tr}}\,\Sigma K^{-1} = \lambda + \frac{\psi\tau\lambda}{\gamma\tau e}\bar{\text{tr}}\,\Sigma(\Sigma + \theta I_d)^{-1} = \lambda + \phi\eta e', \tag{140}$$

$$\tau' = \frac{\lambda}{\tau} = \lambda + \lambda\bar{\text{tr}}\,K_0 K^{-1} = \lambda + \frac{\lambda e}{\gamma\tau e}\bar{\text{tr}}\,\Sigma(\Sigma + \theta I_d)^{-1} = \lambda + (\eta/\gamma)\tau'. \tag{141}$$

We deduce that

$$e' = \frac{\lambda}{1-\phi\eta}, \quad \tau' = \frac{\lambda}{1-\eta/\gamma}, \quad \tau'e' = \lambda\gamma\theta. \tag{142}$$

In particular, the above means that $\eta \le \min(\gamma, 1/\phi)$. The last part of equations (142) can be rewritten as follows

$$\frac{\lambda}{(1-\phi\eta)(1-\eta/\gamma)} = \gamma\theta, \text{ i.e } \phi\eta^2 - (\phi\gamma + 1)\eta + \gamma - \frac{\lambda}{\theta} = 0. \tag{143}$$

This is a quadratic equation for $\eta$ as a function of $\lambda$ and $\theta$, with roots

$$\eta^\pm = \frac{\phi\gamma + 1 \pm \sqrt{(\phi\gamma + 1)^2 - 4(\phi\gamma - (\phi/\theta)\lambda)}}{2\phi} = \frac{\psi + 1 \pm \sqrt{(\psi + 1)^2 - 4(\psi - \phi/\theta')}}{2\phi}. \quad (144)$$

Now, for small $\lambda > 0$ and $\psi \neq 1$, we can do a Taylor expansion to get

$$\eta^\pm \simeq \frac{\psi + 1 \pm |\psi - 1|}{2\phi} \pm \frac{1}{\theta|\psi - 1|}\lambda + O(\lambda^2).$$

More explicitly,

$$\eta^+ \simeq O(\lambda^2) + \begin{cases} 1/\phi + \lambda/((1 - \psi)\theta), & \text{if } \psi < 1, \\ \gamma + \lambda/((\psi - 1)\theta), & \text{if } \psi > 1. \end{cases}$$

$$\eta^- \simeq O(\lambda^2) + \begin{cases} \gamma - \lambda/((1 - \psi)\theta), & \text{if } \psi < 1, \\ 1/\phi - \lambda/((\psi - 1)\theta), & \text{if } \psi > 1, \end{cases}$$

Because $\eta \leq \min(1, 1/\phi, \gamma)$, we must have the expansion

$$\eta \simeq O(\lambda^2) + \begin{cases} \gamma - \lambda/((1 - \psi)\theta), & \text{if } \psi < 1, \\ 1/\phi + \lambda/((\psi - 1)\theta), & \text{if } \psi > 1, \end{cases}$$
$$= \eta_0 - \frac{1}{(1 - \psi)\theta_0}\lambda + O(\lambda^2), \quad (145)$$

provided $\theta_0 > 0$, i.e $\eta_0 \neq 1$. in this regime, we obtain

$$\tau' = \frac{\lambda}{1 - \eta/\gamma} \simeq \begin{cases} \lambda/(1 - 1 + \lambda/((1 - \psi)\gamma\theta_0)) = (1 - \psi)\gamma\theta_0, & \text{if } \psi \leq 1, \\ \lambda/(1 - 1/\psi + o(1)) \to 0, & \text{if } \psi > 1, \end{cases}$$
$$e' = \frac{\lambda}{1 - \phi\eta} \simeq \begin{cases} \lambda/(1 - \psi + o(1)) \to 0, & \text{if } \psi \leq 1, \\ \lambda/(1 - 1 + \lambda\phi/((\psi - 1)\theta_0) \to (\psi - 1)\theta_0/\phi, & \text{if } \psi > 1, \end{cases}$$
$$\tau = 1 - \eta/\gamma \simeq 1 - \eta_0/\gamma = (1 - 1/\psi)_+,$$
$$e = 1 - \phi\eta \simeq 1 - \phi\eta_0 = (1 - \psi)_+.$$

On the other hand, if $\theta_0 = 0$ (which only happens if $\psi < 1$ and $\gamma > 1$ OR $\psi \geq 1$ and $\phi \leq 1$), it is easy to see from (142) that we must have $\tau' \to 0$, $e' \to 0$, $\tau \to 1 - 1/\gamma$, $e \to 1 - \phi \geq 0$.

Next, let's compute the limiting values of $u$ and $\omega' := \omega/\tau^2$. $\qquad\square$

# I   RAW BIAS-VARIANCE DECOMPOSITION

## I.1   CLASSICAL LINEAR MODEL

**Proposition 4.** *Evaluated on the distribution $P_k = P_{\Sigma_k, \sigma_k^2, w_k^*}$, the test error of model $\widehat{f}_{CL}$ defined in (3) is given by*

$$E_{test}(\widehat{f}_{CL}) = B_k + V_k, \quad (146)$$

*where* $V_k = \sum_{j=1}^{2} \frac{\sigma_j^2}{n} r_j^{(4)}(I_d, \Sigma_k),$ $\qquad (147)$

$$B_k = \begin{cases} r_2^{(3)}(\Delta, \Sigma_1) + \lambda^2 r^{(2)}(\Gamma, \Sigma_1), & \text{if } k = 1, \\ r_1^{(3)}(\Delta, \Sigma_2) + \lambda^2 r^{(2)}(\Gamma + \Delta, \Sigma_2) + 2\lambda r_1^{(4)}(\Delta, \Sigma_2), & \text{if } k = 2. \end{cases} \quad (148)$$

*Proof.* Indeed, one computes

$$\mathbb{E}_{\mathcal{D}}\|\widehat{w} - w_k^*\|_{\Sigma_k}^2 = \mathbb{E}_{X_1,Y_1,X_2,Y_2}\mathbb{E}_{x\sim N(0,\Sigma_k)}[(x^\top\widehat{w} - x^\top w_k^*)^2]$$

$$= \mathbb{E}_{X_1,Y_1,X_2,Y_2}\|\widehat{w} - w_k^*\|_{\Sigma_k}^2$$

$$= \mathbb{E}_{X_1,Y_1,X_2,Y_2}\|(M + \lambda I_d)^{-1}X^\top Y/n - w_k^*\|_{\Sigma_k}^2$$

$$= \mathbb{E}_{X_1,Y_1,X_2,Y_2}\|(M + \lambda I_d)^{-1}X^\top(X_1 w_1^* + E_1, X_2 w_2^* + E_2)/n - w_k^*\|_{\Sigma_k}^2$$

$$= \mathbb{E}_{X_1,Y_1,X_2,Y_2}\|(M + \lambda I_d)^{-1}(M_1 w_1^* + M_2 w_2^*) - w_k^*\|_{\Sigma_k}^2 + V_1 + V_2$$

$$= B_k + V_{k,1} + V_{k,2}.$$

where

$$B_k := \mathbb{E}\,\|(M + \lambda I_d)^{-1}(M_k w_k^* + M_{-k} w_{-k}^*) - w_k^*\|_{\Sigma_k}^2, \tag{149}$$

$$V_{k,j} := \frac{\sigma_j^2}{n}\mathbb{E}\,\mathrm{tr}\,M_j(M + \lambda I_d)^{-1}\Sigma_k(M + \lambda I_d)^{-1} = \frac{\sigma_j^2}{n}r_j^{(4)}(I_d, \Sigma_k). \tag{150}$$

It remains to analyze the bias term $B_k$. To this end, observe that

$$(M + \lambda I_d)^{-1}M_k = I_d - (M + \lambda I_d)^{-1}(M_{-k} + \lambda I_d) = I_d - (M + \lambda M)^{-1}M_{-k} - \lambda(M + \lambda I_d)^{-1}.$$

Denoting $M_{-1} = M_2$, $M_{-2} = M_1$, $w_{-1}^* = w_2^*$, $w_{-2}^* = w_1^*$, and $\delta_k = (-1)^k\delta$, where $\delta := w_2^* - w_1^*$, we deduce that

$$(M + \lambda I_d)^{-1}M_k w_k^* + (M + \lambda I_d)^{-1}M_{-k}w_{-k}^* - w_k^*$$

$$= (M + \lambda I_d)^{-1}M_{-k}w_{-k}^* - (M + \lambda I_d)^{-1}M_{-k}w_k^* - \lambda(M + \lambda I_d)^{-1}w_k^*$$

$$= -(M + \lambda I_d)^{-1}M_{-k}\delta_k - \lambda(M + \lambda I_d)^{-1}w_k^\star.$$

Since $w_1^*$ and $\delta_1 = \delta := w_2^* - w_1^*$ are independent with distributions $N(0, \Gamma)$ and $N(0, \Delta)$ respectively, we deduce that

$$B_1 = \|(M + \lambda I_d)^{-1}M_2\delta - \lambda(M + \lambda I_d)^{-1}w_1^\star\|_{\Sigma_1}^2$$

$$= \mathrm{tr}\,\Delta M_2(M + \lambda I_d)^{-1}\Sigma_1(M + \lambda I_d)^{-1}M_2 + \lambda^2\,\mathrm{tr}\,\Gamma_1(M + \lambda I_d)^{-1}\Sigma_1(M + \lambda I_d)^{-1}$$

$$= r_2^{(3)}(\Delta, \Sigma_1) + \lambda^2 r^{(2)}(\Gamma, \Sigma_1).$$

On the other hand, we have $B_2 = B_{2,1} + B_{2,2}$, where

$$B_2 = \|-(M + \lambda I_d)^{-1}M_1\delta - \lambda(M + \lambda I_d)^{-1}w_2^\star\|_{\Sigma_2}^2$$

$$= \|-(M + \lambda I_d)^{-1}M_1\delta - \lambda(M + \lambda I_d)^{-1}(w_1^\star + \delta)\|_{\Sigma_2}^2$$

$$= \|-(M + \lambda I_d)^{-1}(M_1 + \lambda I_d)\delta - \lambda(M + \lambda I_d)^{-1}w_1^\star\|_{\Sigma_2}^2$$

$$= \mathrm{tr}\,\Delta(M_1 + \lambda I_d)(M + \lambda I_d)^{-1}\Sigma_2(M + \lambda I_d)^{-1}(M_1 + \lambda I_d) + \lambda^2\,\mathrm{tr}\,\Gamma(M + \lambda I_d)^{-1}\Sigma_2(M + \lambda I_d)^{-1}$$

$$= \mathrm{tr}\,\Delta M_1(M + \lambda I_d)^{-1}\Sigma_2(M + \lambda I_d)^{-1}M_1 + \lambda^2\,\mathrm{tr}\,\Delta(M + \lambda I_d)^{-1}\Sigma_2(M + \lambda I_d)^{-1}$$

$$\quad + 2\lambda\,\mathrm{tr}\,\Delta M_1(M + \lambda I_d)^{-1}\Sigma_2(M + \lambda I_d)^{-1} + \lambda^2\,\mathrm{tr}\,\Gamma(M + \lambda I_d)^{-1}\Sigma_2(M + \lambda I_d)^{-1}$$

$$= r_1^{(3)}(\Delta, \Sigma_2) + \lambda^2 r^{(2)}(\Gamma + \Delta, \Sigma_2) + 2\lambda r_1^{(4)}(\Delta, \Sigma_2).$$

This completes the proof. $\qquad\square$

## I.2 RANDOM PROJECTIONS MODEL

We now expand the test error $E_{test}(\widehat{f}_{RP})$ of the random projections model $\widehat{f}_{RP}$ defined in (5). For convenience, we recall the definition of the model here. Let $S$ be a $d \times m$ random matrix with iid entries from $N(0, 1/d)$. The model $\widehat{f}_{RP}$ is defined by $\widehat{f}_{RP}(x) := \Phi(x)^\top\widehat{v}$, where $\Phi(x) := S^\top x \in \mathbb{R}^m$ defines a random feature map, and $\widehat{v} \in \mathbb{R}^m$ is given by

$$\arg\min_{v\in\mathbb{R}^m} L(w) = \sum_k \frac{\|\Phi(X_k)v - Y_k\|_2^2}{n} + \lambda\|v\|_2^2. \tag{151}$$

Note that the gradient $\nabla L(v)$ of the regularized loss $L$ is given by

$$\nabla L(v)/2 = \sum_k S^\top X_k^\top (X_k S v - Y_k)/n + \lambda v = \sum_k S^\top M_k S v - \sum_k S^\top X_k^\top Y_k/n + \eta$$

$$= Hv - \sum_k S^\top X_k^\top Y_k/n,$$

where $H := S^\top M S + \lambda I_m \in \mathbb{R}^{m \times m}$, with $M := M_1 + M_2$ and $M_k := X_k^\top X_k/n$. Thus, setting $R := H^{-1}$, we may write

$$\widehat{v} = RS^\top (X_1^\top Y_1 + X_2^\top Y_2)/n = RS^\top (M_1 w_1 + M_2 w_2) + RS^\top X_1^\top E_1/n + RS^\top X_2^\top E_2/n.$$

Now, one deduces the bias-variance decomposition

$$E_{test}(\widehat{f}_{RP}) = \mathbb{E}_{\mathcal{D}} \mathbb{E}_{x \sim N(0,\Sigma_k)}[(\widehat{f}_{RP}(x) - x^\top w_1^*)^2] = \mathbb{E}_{X_1,E_1,X_2,E_2} \|S\widehat{v} - w_k\|_{\Sigma_k}^2 = B_k + V_k,$$

$$\text{where } V_k := V_{k,1} + V_{k,2}, \text{ with } V_{k,j} := \frac{\sigma_j^2}{n} \mathbb{E}_{X_1,X_2} \operatorname{tr} S^\top M_j S R S^\top \Sigma_k S R S^\top,$$

$$B_k := \mathbb{E}_{X_1,X_2} \|SRS^\top (M_1 w_1 + M_2 w_2) - w_k\|_{\Sigma_k}^2.$$

The variance terms $V_{k,j}$ can be directly handled via FPT computations. We now look at the bias term $B_k$. We first treat the case $k = 1$. One has

$$\mathbb{E}\|SRS^\top (M_1 w_1 + M_2 w_2) - w_1\|_\Sigma^2$$
$$= \mathbb{E}\|(SRS^\top (M_1 + M_2) - I_d)w_1 + SRS^\top M_2 \delta\|_\Sigma^2$$
$$= \mathbb{E}\|(SRS^\top M - I_d)w_1\|_\Sigma^2 + \mathbb{E}\|SRS^\top M_2 \delta\|_\Sigma^2$$
$$= \mathbb{E} \operatorname{tr} \Gamma(SRS^\top M - I_d)\Sigma(MSRS^\top - I_d) + \mathbb{E} \operatorname{tr} \Delta M_2 SRS^\top \Sigma SRS^\top M_2$$
$$= \operatorname{tr} \Gamma\Sigma + \operatorname{tr} \Gamma SRS^\top M \Sigma M SRS^\top - 2\mathbb{E} \operatorname{tr} \Gamma\Sigma SRS^\top M + \mathbb{E} \operatorname{tr} \Delta M_2 SRS^\top \Sigma SRS^\top M_2$$
$$= \underbrace{\operatorname{tr} \Gamma\Sigma + \mathbb{E} \operatorname{tr} \Sigma M SRS^\top \Gamma SRS^\top M - 2\mathbb{E} \operatorname{tr} \Gamma\Sigma SRS^\top M}_{\text{classical term } (B)} + \underbrace{\mathbb{E} \operatorname{tr} \Delta M_2 SRS^\top \Sigma SRS^\top M_2}_{\text{extra term } (\zeta)},$$

where we recall that $R := (S^\top M S + \lambda I_m)^{-1}$ and $M := M_1 + M_2$ with $M_k = X_k^\top X_k$.

For the purposes of FPT computations, it might help to observe that $M_k = \lambda \Sigma_k^{1/2} Z_k^\top Z_k \Sigma_k^{1/2}$, where $Z_k := X_k \Sigma_k^{1/2}/(n\lambda)$ is an $n_k \times d$ random matrix with iid entries from $N(0.1/(n\lambda))$. Thus,

$$M_k = \lambda \overline{M}_k, \tag{152}$$

$$\overline{M}_k = \Sigma_k^{1/2} Z_k^\top Z_k \Sigma_k^{1/2}, \tag{153}$$

$$M = \lambda \overline{M}, \tag{154}$$

$$\overline{M} = \overline{M}_1 + \overline{M}_2 = \Sigma_1^{1/2} Z_1^\top Z_1 \Sigma_1^{1/2} + \Sigma_2^{1/2} Z_2^\top Z_2 \Sigma_2^{1/2}), \tag{155}$$

$$R = \overline{R}/\lambda, \tag{156}$$

$$\overline{R} = (S^\top \overline{M} S + I_m)^{-1} = \left( S^\top \Sigma_1^{1/2} Z_1^\top Z_1 \Sigma_1^{1/2} S + S^\top \Sigma_2^{1/2} Z_2^\top Z_2 \Sigma_2^{1/2} S + I_m \right)^{-1}. \tag{157}$$

We need minimal linear pencils for the random matrices

$$A\overline{M}_1 S\overline{R}S^\top B S\overline{R}S^\top, \tag{158}$$

$$A\overline{M} S\overline{R}S^\top B S\overline{R}S^\top \overline{M} \tag{159}$$

$$AS\overline{R}S^\top \overline{M}, \tag{160}$$

$$A\overline{M}_2 S\overline{R}S^\top B S\overline{R}S^\top \overline{M}_2, \tag{161}$$

in terms of the set of free variables $\{A, B, \Sigma_1^{1/2}, \Sigma_2^{1/2}, S, Z_1, Z_2, S^\top, Z_1^\top, Z_2^\top\}$. Observe that

$$\operatorname{tr} AMSRS^\top BSRS^\top M$$
$$= \operatorname{tr} AM_1 SRS^\top BSRS^\top M_1 + \operatorname{tr} AM_2 SRS^\top BSRS^\top M_2 + 2\operatorname{tr} AMSRS^\top BSRS^\top M,$$
$$\operatorname{tr} ASRS^\top M = \operatorname{tr} ASRS^\top M_1 + \operatorname{tr} ASRS^\top M_2.$$

For our business, it is therefore sufficient to only compute (minimal) linear pencils for

$$AS\overline{R}S^\top \overline{M}_1, \tag{162}$$
$$A\overline{M}_1 S\overline{R}S^\top BS\overline{R}S^\top, \tag{163}$$
$$A\overline{M}_1 S\overline{R}S^\top BS\overline{R}S^\top \overline{M}_1, \tag{164}$$
$$A\overline{M}_1 S\overline{R}S^\top BS\overline{R}S^\top \overline{M}_2, \tag{165}$$

where $\overline{M}_k := \Sigma_k^{1/2} Z_k^\top Z_k \Sigma_k^{1/2}$, $\overline{R} := \left(S^\top \overline{M} S + I_m\right)^{-1}$, $\overline{M} := \overline{M}_1 + \overline{M}_2$.

Observe that without the $S$ matrix (i.e taking $m = d$ and $S = I_d$), the four matrix expressions above reduce to what we had in the classical case.

