# OpenReview forum: "Strong Model Collapse"
_ICLR.cc/2025/Conference — ICLR 2025 Spotlight_

### Official Review · Reviewer_g7eD · 2024-10-23

**Soundness:** 4
**Presentation:** 4
**Contribution:** 3
**Rating:** 8
**Confidence:** 4

**Summary:**

This paper studies the model collapse phenomenon in a supervised regression setting. The authors study whether model collapse is inevitable or whether proper mixing strategies can mitigate it. They also study the connection between model size and robustness to model collapse.

*Setup and tools*: The theoretical analysis focuses on linear models and random projection models (simplification of neural networks). The authors derive their proofs using linear pencils and operator-valued free probability theory (OVFPT).

*Contributions*: The authors provide two main theoretical results: 1) they show that even the smallest fraction of synthetic data can lead to model collapses where increasing the model's size does not improve the performance. This is dubbed "strong model collapse". 2) They study the impact of the model's size on the severity of the collapse and show that bigger models are more prone to model collapse when the distribution shift between synthetic and real data is important. They also show that an interpolation threshold exists (model size = sample size) beyond which larger models are more robust to collapse. This is akin to a double descent phenomenon. The theoretical results are experimentally verified on synthetic data and the authors also conduct experiments with neural networks on MNIST and GPT2-small on BabyStories data that corroborate their results.

**Strengths:**

- The problem tackled by the authors is of great interest for the training of large-scale generative models.
- I find the derivation of the proof elegant and the usage of the OVFPT novel in this setting.
- The derived results are insightful and well-analyzed by the authors, along with the empirical validation on synthetic data that helps to convey their implications to real-world scenarios.
- The authors conduct experiments on real data with (small) neural networks and LLMs that corroborates their theoretical analysis.
- The paper is well-written and logically structured, making it easy to follow the authors' arguments and methodology.

**Weaknesses:**

1) Could the authors detail more the key steps/ideas of the proofs in the main paper (paragraph **Proving Theorem 1**)? I believe such tools could be beneficial to the reader, even in other applications.
2) Could the authors discuss the estimation of the quality of synthetic data (parameter $c^2$) on the real applications with MNIST and BabiStories? How would one assess it when training a large-scale model like an LLM  or a VLM?
3) Could the authors discuss in more length the "cost" of approximating neural networks as random projections? In particular, what is lost in terms of generality, does it approximate any architecture (e.g., can LLMs be studied in the NTK or infinite-width limit?)

**Questions:**

See weaknesses.

I list the potential typos found in the text below. *"a-->b"* means I think a should be replaced by b.

*Section 2.2*
- Paragraph **Random Projection Models** (l224 & l225 ): it seems that dimensions of $S, x, v$ do not match. According to previous notations, $x \in \mathbb{R}^d$ implies $S \in \mathbb{R}^{m \times d}$ (instead of $d \times m$ as it is currently (l224)) and $v \in \mathbb{R}^m$ (instead of $v \in \mathbb{R}^k$ as it is currently (l225)).

*Section 3.1*
- Paragraph **Example: The Isotropic Case** (l299): *"the synthetic data $P_1$"* --> it should be *"the synthetic data $P_2$"*

*Section 3.2*
- Paragraph **Are Larger Models More Prone...** (l398 & l401): *"represented by crosses"* --> it should be *"represented by stars"* (l398). I am not sure but according to Figure 1 *"for both high-quality and low-quality (squares)"* --> it should be *"for both very high-quality and high-quality (squares and diamonds)"* (l401)

---

> ### Author Response · Authors · 2024-11-19
> **Author Response**
>
> We thank the reviewer for their positive reception of our work and their appreciation of the importance of the topic and the elegance of our approach. We will now address their questions and comments.
>
>
> >*Could the authors detail more the key steps/ideas of the proofs in the main paper (paragraph Proving Theorem 1)? I believe such tools could be beneficial to the reader, even in other applications.*
>
> Sure. We have updated the manuscript with a modified version of that paragraph (see text in blue) which explains the key steps of the proof. Note that proofs of all results are given in the appendix. Also, there is a detailed ToC in the beginning of the appendix to help the reader navigate the proofs.
>
> >*Could the authors discuss the estimation of the quality of synthetic data (parameter ) on the real applications with MNIST and BabiStories? How would one assess it when training a large-scale model like an LLM or a VLM?*
>
> For the quality of the MNIST and BabiStories datasets, we directly assess the accuracy or perplexity of the generator to evaluate the quality of the synthetic data, as also noted by Reviewer Wc8A. Additionally, we have included an ablation study in Appendix A.3 to examine how the quality of synthetic data impacts the scaling curves.
>
> For general pretraining data for LLMs and VLMs, evaluating data quality is a broad and evolving area. Researchers have proposed various metrics in natural language processing and have utilized scores from pretrained models, such as the CLIP score [1], to evaluate image-caption pairs. Scaling these methods to assess large, general-purpose corpora remains an open challenge and an active area of research.
>
> [1] Radford, Alec, et al. "Learning transferable visual models from natural language supervision." International conference on machine learning. PMLR, 2021.
>
> >*Could the authors discuss in more length the "cost" of approximating neural networks as random projections? In particular, what is lost in terms of generality, does it approximate any architecture (e.g., can LLMs be studied in the NTK or infinite-width limit?)*
>
> For the purpose of studying the impact of model size on the model collapse phenomenon, we choose to analyze ridge regression with random projections because this setup offers analytical tractability while producing a phenomenology which is strictly richer than in the case of classical linear models (with no random projections).  Furthermore, still using free probability theory and the Gaussian equivalence theorem [Goldt et al. (2022)], we can easily extend our theory to allow for the nonlinearities and other complexities in the NTK regime, as in (Adlam & Pennington, 2020; Tripuraneni et al., 2021) in studying other phenomena like multiple descent and covariate shift, albeit at the cost of more complicated fixed-point equations describing the limiting value of the test error.
>
> The question of whether NTK can approximate LLMs is broad with no simple answer. In any case, we can say the following (1) such an NTK would have to be finite-width, since infinite-width NTK's cannot account for feature learning (which is definitely present in LLMs). (2) In the so-called "lazy regime", such an approximation is plausible, due to the universality established in (Chizat et al., 2019).
>
>
> **Typos.** We also thank the reviewer for the typos they have pointed out.
>
> >*Paragraph Random Projection Models (L224 & L225 ): it seems that dimensions of $S$, $x$, $v$ do not match*
>
> Indeed, there is a mild typo (missing transpose on $S$): $Sx$ should be $S^\top x \in \mathbb R^m$, with $x \in \mathbb R^d$, $S \in \mathbb R^{d \times m}$, and $v \in \mathbb R^m$. This has been fixed (see blue text in updated manuscript).
>
> We also thank the reviewer for the other typos they have pointed out (for example, the first "$P_1$" should indeed be "$P_2$" on L299; etc.). They have all been fixed.

---

> > ### Comment · Reviewer_g7eD · 2024-11-20
> > **Answer to Authors**
> >
> > I thank the authors for their detailed answers which address my concerns. I appreciate the additional experiments and discussion in the revised manuscript.
> >
> > I maintain my (already high) score as, to me, this is very qualitative work that deserves acceptance.

---

### Official Review · Reviewer_5qx9 · 2024-11-02

**Soundness:** 3
**Presentation:** 3
**Contribution:** 3
**Rating:** 8
**Confidence:** 3

**Summary:**

This paper considers the questions: is model collapse inevitable when training with synthetic data? Are larger models more prone to collapse than smaller ones? The paper concludes that model collapse is inevitable if training with some non-zero about of synthetic data. Larger models may suffer more from model collapse in the low quality data setting. The paper contains theoretical analysis for linear models and random projection models, and empirical results for GPT-2 models trained on the BabiStories dataset.

**Strengths:**

- The paper shows model collapse in theoretical settings of the linear and random projection models
- The paper shows nice alignment between the experiments and theoretical analysis (eg Fig 3)

**Weaknesses:**

I'm a bit unclear on the takeaway from the experimental results with GPT2. Specifically, the paper seems to make mixed claims for the case of large models beyond the interpolation threshold; it is said that large models may mitigate the collapse beyond the interpolation threshold and that large models tend to amplify collapse beyond the interpolation threshold in the experimental results.

**Questions:**

- For the empirical results, it is said that the curves are expected to plateau in Fig 5. Can this be shown empirically by further scaling?
- Can the double descent phenomenon be showed more clearly in the experimental results?
- Separately, I would be interested in more analysis specifically in the high quality data setting, ie what can be said about the amount of model collapse as the synthetic distribution approaches the real distribution?

---

> ### Author Response · Authors · 2024-11-19
> **Author Response**
>
> We thank the reviewer for their positive reception of our work. We will now address their questions and comments.
>
> >*I'm a bit unclear on the takeaway from the experimental results with GPT2. Specifically, the paper seems to make mixed claims for the case of large models beyond the interpolation threshold; it is said that large models may mitigate the collapse beyond the interpolation threshold and that large models tend to amplify collapse beyond the interpolation threshold in the experimental results.*
>
> The term "beyond the interpolation threshold" refers to increasing the network size beyond the point where a fixed dataset can be fully interpolated. In the GPT-2 experiment, however, we vary the number of tokens. As the number of tokens increases, the setting transitions from interpolating all data to not interpolating. Beyond the interpolation threshold corresponds to scenarios with fewer tokens, where a larger model mitigates collapse and achieves better performance. Conversely, when the parameter size is insufficient, as shown in the right end of Figure 5 (right), larger models tend to amplify collapse.
>
> >*For the empirical results, it is said that the curves are expected to plateau in Fig 5. Can this be shown empirically by further scaling?*
>
> Theoretical predictions suggest that the scaling curves will plateau. In the empirical setting, with the x-axis now on a logarithmic scale, extending the range slightly already doubles the already high training requirements of a pretraining task, making the demands even more substantial. Moreover, to properly demonstrate scaling law curves for a long range, it would be necessary to further increase the model size when the training token count increases, significantly raising computational costs. Unfortunately, these experiments are too expensive and resource-intensive for us to conduct as part of our rebuttal (it would take more than 2 weeks running on 8 V100s).
>
> Our theory predicts that optimal performance is bounded by a constant that depends on $c^2$ (data quality) and $p_2$, the proportion of synthetic data. The case of small $p_1$ is already demonstrated in Figure 5 (Right), and we expect similar trends to hold when extrapolating the other curves in Figure 5 (Left).
>
> >*Can the double descent phenomenon be showed more clearly in the experimental results?*
>
> We clearly demonstrate the double descent curve in Figure 8 in Appendix A.2 using the MNIST dataset. As the number of samples increases, the curves exhibit a characteristic "down-and-up" pattern. Notably, when the sample size is fixed, the performance rankings of models with different parameter sizes change depending on the sample size, further illustrating the double descent phenomenon.
>
> In the current Figures 10 and 11, provided in response to Reviewer Wc8A, we observe the double descent behavior more explicitly when increasing the model size while keeping the sample size fixed. This closely mirrors the theoretical double descent curve shown in Figure 4. With different combinations of the number of synthetic data $n_2$ and hidden dimension $d$, the figure captures various segments of the double descent curve.
>
> In Figure 5, the double descent behavior can also be interpreted through the changing performance rankings of models with different sizes as the sample size varies.
>
> >*Separately, I would be interested in more analysis specifically in the high quality data setting, ie what can be said about the amount of model collapse as the synthetic distribution approaches the real distribution?*
>
> In our theoretical framework, high-quality synthetic data corresponds to small values of the quality $c^2$ parameter (smaller is better; refer to **Definition 1**). The amount of model collapse  is a decreasing function of $c^2$. Letting $c^2$ tend to zero (i.e synthetic data with quality tending to perfection) in **Theorem 1** and **Theorem 2** and their corollaries (e.g **Corollary 1** of **Theorem 1**) recovers classical results. Of course, this limit also matches $p_2 \to 0^+$, corresponding to the setup where all training data is from the true data distribution. Our results show that for **fixed** positive values of $c^2$ and $p_2$ however small, model collapse persists. This is what we refer to as *strong* model collapse.
>
> We hope that we addressed all your concerns, and that you are satisfied with our responses and changes. If so, please consider raising your score.

---

### Official Review · Reviewer_Wc8A · 2024-11-06

**Soundness:** 3
**Presentation:** 3
**Contribution:** 3
**Rating:** 8
**Confidence:** 3

**Summary:**

This paper studies model collapse, which relates the performance degradation of ML models due to synthetic training data. For linear regression and random projections model, the authors establish scaling laws that characterize this degradation, as a function of the proportion of the synthetic data as well as the quality. Their findings suggest that larger models become more susceptible to this degradation. Experiments pertaining to training GPT-2 with synthetic data are used to demonstrate validity of their results beyond the limited theoretical models.

**Strengths:**

- The paper is very well written; the problem statement and the contributions are clearly established.
- The theoretical results seem sound and intriguing, and I appreciate the authors efforts to present simplified examples to help the reader understand the implications better.
- The findings seem to transfer to practical learning settings fairly well.

**Weaknesses:**

- Model Assumptions: I am unsure of the modelling of synthetic data and how well it translates into practice. Specifically, the authors model synthetic data using a label shift, assuming that the data (X) marginal remains the same. However, it seems unrealistic for autoregressive training (a key experiment in the paper), where the input tokens for next token generation come from the synthetic distribution.
- Experimental Details: The theoretical results establish a strong dependence on the quality of synthetic data. However, the experiments with real data (MNIST/GPT-2) do not provide quantitative metrics to measure the degradation in the synthetic data source (either accuracy of MNIST classifier or perplexity/goodness scores of the trained GPT-2 generator), which makes it hard to ascertain which paradigm (in fig.1) the experiments align best with or what level of degradation in practice results in the observed trend.
- Minor Issues
    - Typos: Line 225 (v \in R^{m}), line 299 (synthetic data P2), line 398 (represented by stars)
    - Suggestions for Clarity:
        - Akin to theorem 1, is it possible to present a simplified version of theorem 2 for the general audience? As it is, definition 2 and theorem 2 are hard to digest just on their own.
        - Line 481, before stating the result, can the authors explain in words the process of iterative mixing proposed in Ferbach et al., 2024? It would make the manuscript more self-contained.
    - Missing Citation: Line 424, for the MNIST dataset, please include the citation for "Gradient Based Learning Applied to Document
 Recognition", LeCun et al, 1998
    - Visualization: For fig.1, please consider changing the y-axes test error to the same scale. Right now, it is hard to compare the error values or the slope in subplot 1 to those in 2 and 3.

**Questions:**

- Model Assumptions: Regarding the label shift assumption, what are the authors thoughts on how the results would change if the domain shift were to be accounted for as well?
- Experimental Setup:
    - I understand that it might not be computationally feasible for GPT-2 in the given timeframe, but could the authors provide an ablation on the MNIST experiment with regards to the quality of synthetic data? That is, models trained on labels taken from increasingly worse synthetic classifier and the corresponding trends.
    - Minor: Regarding the GPT-2 experiment, if BabiStories is itself synthetic, why not use TinyStories as real data and BabiStories as the synthetic dataset for training the models, as opposed to creating a new GPT generator?

---

> ### Author Response · Authors · 2024-11-19
> **Author Response (Part 1)**
>
> We thank the reviewer for their time and effort, and their positive appreciation of our theoretical contributions and illustrative experiments. We will now address their questions and comments.
>
> >*Model Assumptions: I am unsure of the modelling of synthetic data and how well it translates into practice. Specifically, the authors model synthetic data using a label shift, assuming that the data (X) marginal remains the same. However, it seems unrealistic for autoregressive training (a key experiment in the paper), where the input tokens for next token generation come from the synthetic distribution.*
>
> **Label-shift and covariate shift.** At the cost of much more cumbersome theorems and formulae, theoretical results in our paper can be easily extended to accommodate scenarios where the covariance matrix of the synthetic data distribution $\Sigma_2$ is different from the covariance matrix of the true data distribution $\Sigma_2$, alongside the label-shift setup we used in our work. It is a matter of invoking the general bias-variance decompositions established in **Appendix I** and combining with the deterministic equivalents provided in **Appendix E**. We chose to keep things simple.
>
> Let us note that an assumption like $\Sigma_1 \ne \Sigma_2$ alone (i.e without label-shift) corresponds to *covariate-shift*, a setup studied extensively in the literature Shimodaira (2000); Tripuraneni et al., 2021; (also see the classical work [1] listed below), but is not sufficient to capture model collapse as understood in the literature. Our work focuses on label-shift by choice. Let us recall here that the main payload of our work are negative results which indicate that the model collapse phenomenon persists even when synthetic and real data are mixed, showing that the findings of Shumailov et al. (2024); Alemohammad et al.
> (2023); Bertrand et al. (2023); Dohmatob et al. (2024a;b) are worse than anticipated. In line with prior work, the setup of label-shift is sufficient for highlighting our striking result, while keeping theorems as simple as possible. We will make these points clearer in the manuscript.
>
> [1] Shai Ben-David et al. "Impossibility Theorems for Domain Adaptation", AISTATS, 2010.
>
> **Auto-regressive models.** We adopt the setting of X-Y pairs theoretically, as it encompasses a broad application range in language modeling, particularly in post-training scenarios where the model receives a prompt (X) and generates a response (Y). This setup includes instruction tuning, standard fine-tuning, and tasks such as question answering and code generation.
>
> We deliberately chose the GPT-2 autoregressive experiment because it is a pretraining tasks, where model collapse is claimed to have a significant impact (Shumailov et al. (2024); Dohmatob et al. (2024a)), rather than fine-tuning. Many related works conduct experiments exclusively on fine-tuning outcomes. In the current story dataset, prompts and responses also play a role in dataset curation: a set of prompts was first created and then used to query GPT-4 (Mistral-8x7B) and our trained GPT-2 to generate BabyStories (TinyStories) and the synthetic data. The dataset is subsequently utilized in the autoregressive pretraining process. The experimental results suggest that insights derived from the prompt-response setting can effectively generalize to pretraining tasks.
>
> >*I understand that it might not be computationally feasible for GPT-2 in the given timeframe, but could the authors provide an ablation on the MNIST experiment with regards to the quality of synthetic data? That is, models trained on labels taken from increasingly worse synthetic classifier and the corresponding trends.*
>
> We have included the results of additional experiments on MNIST in Figures 10 and 11 in Appendix A.3. These figures complement Figure 8. In these experiments, we examine how the quality of synthetic data influences the relationship between the number of synthetic data points, model size, and performance. The quality of synthetic data is evaluated using the MSE loss of the generator on the test set, consistent with our approach of framing the classification problem as a regression task. For the same dataset and trained model, we also use test accuracy to evaluate the quality and performance in Figure 11. We also keep the y-axis in the same scale.

---

> > ### Author Response · Authors · 2024-11-19
> > **Author Response (Part 2)**
> >
> > In the figures, the synthetic data with the highest quality already achieves accuracy close to the optimal. As the quality decreases, we observe that the shape of the curve begins to resemble a double descent curve, similar to the changes in the Pareto frontiers shown in Figure 1. With different combinations of the number of synthetic data $n_2$ and hidden dimension $d$, the figure captures various segments of the double descent curve depicted in Figure 4. When $n_2$ is small (as seen in the left subplots), it corresponds to a large parameterization rate $\psi$, placing it in the second descent region of the double descent curve. Conversely, when $n_2$ is large (as shown in the right subplots), it captures the up-and-down behavior characteristic of the double descent phenomenon.
> >
> > >*Visualization: For fig.1, please consider changing the y-axes test error to the same scale. Right now, it is hard to compare the error values or the slope in subplot 1 to those in 2 and 3*
> >
> > We deliberately presented the results this way because the goal is not to compare the results for different values of $n_2$, but to highlight the impact of synthetic data quality and model size on model collapse. We have followed the reviewer's advice and updated the figure to show all y-axes on the same scale. Please see the updated version of the manuscript.
> >
> > >*Line 481, before stating the result, can the authors explain in words the process of iterative mixing proposed in Ferbach et al., 2024? It would make the manuscript more self-contained.*
> >
> > In the iterative mixing approach proposed in Ferbach et al. (2024), a model is fitted on a mixture of synthetic and real data. In the next iteration, the labels for the synthetic data are replaced with the labels predicted by the previous iteration. The process continues for $t$ iterations. We have updated the manuscript with explicit description in plain english as suggested by the reviewer.
> >
> >
> > >*Akin to theorem 1, is it possible to present a simplified version of theorem 2 for the general audience? As it is, definition 2 and theorem 2 are hard to digest just on their own.*
> >
> > We agree with the reviewer that definitions and theoretical results should be unpacked and provided in simplified version whenever possible, as was done in **Definition 1**, **Theorem 1**, and corollaries. Therefore in the spirit of **Theorem 1** and corollaries (which handle classical regression), the rest of the section just after the statement of **Theorem 2** is spent unpacking **Theorem 2**; we isolate the effect of synthetic data quality $c^2$ and synthetic data proportion $p_2$. Other important effects (e.g model size) are implicitly burried in the constants $e$, $\tau$, $u$, $\theta$, and $\omega$ defined in **Definition 2**.
> >
> > The scalar $u$ is a kind of pseudo-variance term; in fact it can be easily shown that the variance term ($V$) in the bias-variance decomposition becomes $\sigma^2 u$ in the limit $\psi \to \infty$ (corresponding to extremely over-parametrized random projections), matching the formula given in Theorem 1 for classical linear regression.
> >
> > Since the other constants constants ($e$, $\tau$, $\theta$, and $\omega$) are defined implicitly via generally complicated scalar fixed-point equations, its is difficult to give a simple description thereof like was done in **Theorem 1** and **Definition 1**. Notwithstanding, in **Appendix H** We explicitly compute these constants analytically in the unregularized limit $\lambda \to 0^+$.
> >
> > **Typos.** We correct all the typos and also follow the suggestions of the reviewer regarding the presentation of some of the plots in our manuscript.

---

> > > ### Author Response · Authors · 2024-11-19
> > > **Author Response (Part 3)**
> > >
> > > > Minor: Regarding the GPT-2 experiment, if BabiStories is itself synthetic, why not use TinyStories as real data and BabiStories as the synthetic dataset for training the models, as opposed to creating a new GPT generator?
> > >
> > > TinyStories and BabiStories are both extremely high-quality datasets, created using GPT-4 and Mistral-8x7B, respectively, and can both be considered real data. For our experiments, we use BabiStories as the real dataset and a GPT model pretrained on this dataset (BabiStories) as the synthetic data generator. This setup mirrors the model collapse settings described in prior work (Shumailov et al., 2024), where the synthetic data generator is pretrained on the real dataset (BabiStories) we have.
> > >
> > > Additionally, this design takes into account the generation costs for obtaining empirical scaling curves. The generation cost of GPT-4 and Mistral-8x7B is prohibitively high, making the GPT-2 generator a more practical choice for producing larger datasets to support a broader range of scaling experiments. TinyStories was not used due to license restrictions associated with one of the author's affiliations.
> > >
> > > We hope our additional explanations and the experiments we have now performed upon your suggestion clarify things. We would be grateful if you could let us know what you find lacking and raise your score if this finds your approval.

---

> > ### Comment · Reviewer_Wc8A · 2024-11-22
> > **Regarding Model Assumptions**
> >
> > - Label-shift and covariate shift: Regarding modeling assumptions, my question was whether label-shift on its own captures the gist of autoregressive training experiment. Specifically, consider the sequence of n tokens $X_{[1:m]} X_{[m+1: n]}'$, which represents the good prompt (m tokens from X) and the story generated by the GPT-2 model (n-m tokens from X'). Then the loss term for token $X_i',  (i \in \{m+1, ....., n\})$ is computed as a function of the input sequence $X_{[1:m]} X_{[m+1:i-1]}'$, which represents a weird mix of the both the real and synthetic data distribution for the input (domain shift) as well as the output token (label shift). Hence, my concern of whether results in label shift alone directly transfer to this setting. In that regard, I am satisfied to hear that the results can indeed be extended to a reasonable extent.

---

> > > ### Comment · Reviewer_Wc8A · 2024-11-22
> > > **General Response to Rebuttal**
> > >
> > > The authors have addressed most of my concerns to a reasonable extent and the proposed changes make for a better reading. Thus, I am happy to increase my score from 6 to 8.

---

> > > > ### Author Response · Authors · 2024-11-22
> > > > **Final Comment**
> > > >
> > > > We thank reviewer for their insightful comments and kind suggestions which have greatly helped improve the presentation of our work. We also thank the reviewer for updating their score.

---

### Author Response · Authors · 2024-11-19
**General Response**

We thank all the reviewers for their overall positive reception of our work. We are also grateful for their thoughtful and helpful feedback.

In this rebuttal, we carefully address all the questions and comments of the reviewers. We have also run additional experiments to address specific requests from the reviewers, to help clarify some of our results.

Our rebuttal comes alongside an updated version of the manuscript, that includes new figures and text changes (in blue).

---

### Public Comment · ~Rylan_Schaeffer2 · 2024-11-30
**Scientific Claims Require Discussion & Citation of Intentionally-Omitted Contradictory Prior Work**

**TLDR: The authors' decision to not cite our directly relevant published work - of which they are well aware through significant interactions - while using our methodology and reaching contradictory conclusions without explanation, warrants careful examination.**

“Strong Model Collapse” reaches conclusions that appear to be in tension with recent and relevant work, particularly [Gerstgrasser et al. 2024's "Is Model Collapse Inevitable? Breaking the Curse of Recursion by Accumulating Real and Synthetic Data," published at COLM 2024](https://openreview.net/forum?id=5B2K4LRgmz#discussion). However, while the authors of “Strong Model Collapse” are well aware of “Is Model Collapse Inevitable?” and even use its methodology, the authors do not cite the work.

Note: I am a co-first author of “Is Model Collapse Inevitable?”

**Scientific Contradictions with Prior Work:**  The scientific claims of “Strong Model Collapse” merit further discussion in light of existing work. Specifically, “Strong Model Collapse” concludes that model collapse "generally persists even when mixing real and synthetic data, as long as the fraction of training data which is synthetic does not vanish” and that "model collapse cannot generally be mitigated by simple adjustments such as data weighting unless these strategies asymptotically remove all but a vanishing proportion of synthetic data from the training process.” However, our work and other existing literature (including other manuscripts by the authors of “Strong Model Collapse”) demonstrate that model collapse can be robustly avoided, even when the fraction of real data vanishes - an apparent contradiction that warrants careful examination.

**Uncited Reuse of Experimental Methodology:** The experimental setting of pretraining sequences of GPT-2 causal transformers on TinyStories was first used in “Is Model Collapse Inevitable” and is now re-used in “Strong Model Collapse” without citation. While perhaps conceptually straightforward, finding the correct experimental combination of models, data and optimization that faithfully emulates model collapse in the face of high costs from pretraining sequences of language models was non-trivial. “Is Model Collapse Inevitable” also contributed results in the linear regression setting, the same setting considered in Section 2 of “Strong Model Collapse,” that similarly go uncited.

**The Authors Acted Intentionally:** While missing a citation is in general an innocuous and easily corrected oversight, in this particular case, the authors of “Strong Model Collapse” had direct engagement with “Is Model Collapse Inevitable” beginning in April 2024, including (i) providing feedback that was incorporated into the April 29th 2024 Arxiv update and (ii) publicly tweeting about our manuscript over summer.

**Conclusion:** We respectfully ask the reviewers and the AC to ask the authors of “Strong Model Collapse” to address:

1. The relationship between this work and existing literature on avoiding model collapse, and

2. How the seemingly contradictory conclusions arise from different modeling assumptions, and which assumptions best reflect real-world scenarios.

Evaluating which assumptions are most relevant to real-world scenarios is essential for assessing the practical implications of this work.

Note: I have omitted additional relevant evidence to preserve anonymity of the authors. If the AC wishes to see said evidence, I can provide it.

---

> ### Author Response · Authors · 2024-12-01
> **Response to concerns (part 1)**
>
> ## Point 1. Concerning Gerstgrasser et al. (2024)
>
> > ***Scientific Contradictions with Prior Work ...***
>
> This statement from the commenter is inaccurate and misleading. Indeed the commenter's own work Gerstgrasser et al. (2024) (mentioned in their comment) doesn't avoid model collapse in any meaningful way. Let us explain.
>
> In our introduction, we clearly define model collapse as "a critical degradation in the performance of AI models." **Avoiding model collapse would mean to *close the performance gap between training with real and synthetic data.***
>
> The definition adopted in the commenter's own work Gerstgrasser et al. (2024) describes avoiding model collapse as preventing recursive degradation when models are trained multiple times. The paper acknowledges that, while performance degradation is bounded when samples accumulate, there is still a performance loss. This definition represents only a partial condition for closing the gap between real and synthetic data. In this sense, Gerstgrasser et al. (2024) doesn't solve or mitigate model collapse.
>
> In the broader literature on model collapse, the prevailing trend considers closing the performance gap as the primary criterion for avoiding model collapse. We chose to align with this widely adopted approach.
>
> From a practitioner's perspective, closing the gap between real and synthetic data is a more actionable and relevant notion of avoiding model collapse. Simply ensuring non-diverging performance can still result in models failing to match the synthetic data generator's quality, thereby causing synthetic data to harm performance. Only when the gap is fully closed can the harm from synthetic data be entirely mitigated. We believe this principle should guide practitioners.
>
> Due to the differences in definitions and the rationale for considering closing the gap as the correct definition, we have reviewed all related work on understanding model collapse and the strategies to avoid it through this widely accepted perspective.
>
> Additionally, the referenced paper Gerstgrasser et al. (2024) was published at COLM (October 2024), after the ICLR submission deadline, with its camera-ready deadline in August. According to the ICLR author guidelines, papers published after July 1st are considered contemporary work.
>
> **Prior interaction with the authors of Gerstgrasser et al. (2024)**
>
> Unfortunately, one of the statements of the comment poster seems to break anonymity of this review process. We did have an interaction with the authors of Gerstgrasser et al. (2024) months ago when their paper first appeared on ArXiv. Let us succinctly summarize the conclusions of the said interaction:
>
> - **Technically insufficient.** The technical contribution of Gerstgrasser et al. (2024) was too incremental and very weak. It was nothing more than a slight modification of one of the setups and arguments already presented in an earlier version of Dohmatob et al. (2024a) "Model Collapse Demystified: ...". Their results can be recovered as a trivial corollary of Theorem 4.1 of Dohmatob et al. (2024a). Also see **Remark 4.2** of Dohmatob et al. (2024a).
> - **Misleading/inaccurate conclusions.** The paper simply didn't solve model collapse in any reasonable sense (see our previous note on definition, further above, and also the **related work** sections of this ICLR submission), contrary to the claims made by Gerstgrasser et al. (2024).
>
> Unfortunately, later versions of the paper Gerstgrasser et al. (2024) essentially ignored the constructive criticisms we kindly shared during this interaction, relegating our remarks to a footnote at the end of the paper. Therefore in its current state, we still don't think Gerstgrasser et al. (2024) is scientifically sound (for example, in terms of the technical contributions and most importantly, the misleading/inaccurate claims of solving/mitigating model collapse, as explained above), and therefore didn't feel the need to cite.

---

> ### Author Response · Authors · 2024-12-01
> **Response to concerns (part 2)**
>
> ## Point 2. Related Work on Avoiding Model Collapse
>
> The statement from the commenter regarding "simple adjustments such as data weighting unless these strategies asymptotically remove all but a vanishing proportion of synthetic data from the training process" is a faithful interpretation of our theoretical results. To the best of our knowledge, this characterization does not contradict related works in the field with the same definition of avoiding model collapse. The claim is firmly grounded in theory, and our theoretical framework has been rigorously vetted by reviewers. Specifically, we examine *Strategic Data Mixing* with data source coefficients for data mixing and demonstrate that it will not prevent model collapse unless the strategic weights of synthetic data asymptotically vanish.
>
>
> ## Point 3. Reuse of Experimental Methodology
>
> Our choice to use GPT-2 and the BabiStories setting is driven by several factors: its role as a pretraining task, its ability to generate more 'real' data to support the scaling analysis shown in Figure 5 (Left), and the availability of comprehensive resources, including generation code, checkpoints, and training code, provided in [1]. These resources, combined with the use of Mistral to generate additional data for BabiStories, enabled us to successfully conduct the large-scale scaling experiments detailed in Figure 5.
>
> The trained GPT-2 checkpoints, generated samples, training code, and generation code were obtained from the public GitHub repository of the Memory Mosaics paper [1]. We extended this codebase to perform our experiments and are fully prepared to provide evidence of this to the AC and reviewers if needed.
>
> Furthermore, we emphasize that TinyStories and BabiStories have become widely recognized as standard dataset for affordable pretraining [1]. These datasets are valued for their efficiency in facilitating the pretraining of high-performing language models at moderate scales, as evidenced by numerous studies [2–10]. We thus emphatically refute the claim that we have used experiments from Gerstgrasser et al without acknowledgement - or else the same is true for Refs [1-10] below.
>
> [1] Zhang, Jianyu, et al. "Memory Mosaics." arXiv preprint arXiv:2405.06394 (2024).
>
> [2] Raises, Lying Leg, and Side Plank. "How to Generate and Use Synthetic Data for Finetuning."
>
> [3] Cuervo, Santiago, and Ricard Marxer. "Scaling Properties of Speech Language Models." arXiv preprint arXiv:2404.00685 (2024).
>
> [4] Radhakrishnan, Adityanarayanan, et al. "Mechanism for feature learning in neural networks and backpropagation-free machine learning models." Science 383.6690 (2024): 1461-1467.
>
> [5] Gu, Yuxian, et al. "Towards Optimal Learning of Language Models." arXiv preprint arXiv:2402.17759 (2024).
>
> [6] Shah, Harshay, Andrew Ilyas, and Aleksander Madry. "Decomposing and editing predictions by modeling model computation." arXiv preprint arXiv:2404.11534 (2024).
>
> [7] Razzhigaev, Anton, et al. "Your Transformer is Secretly Linear." arXiv preprint arXiv:2405.12250 (2024).
>
> [8] Sharifnassab, Arsalan, et al. "Soft Preference Optimization: Aligning Language Models to Expert Distributions." arXiv preprint arXiv:2405.00747 (2024).
>
> [9] Zhao, Yize, et al. "Implicit geometry of next-token prediction: From language sparsity patterns to model representations." arXiv preprint arXiv:2408.15417 (2024).
>
> [10] Zhong, Ziqian, and Jacob Andreas. "Algorithmic Capabilities of Random Transformers." arXiv preprint arXiv:2410.04368 (2024).

---

> ### Author Response · Authors · 2024-12-01
> **Response to concerns (part 3/3)**
>
> **Final Note.** The perceived contradictions in conclusions primarily arise from differing definitions of what constitutes avoiding model collapse, as detailed in our response to point **1** above. These differences have no bearing on the underlying modelling assumptions (as claimed by the commenter), but rather arises from a misguided view of the model collapse phenomenon in their own paper Gerstgrasser et al. (2024). The perspective and findings of our work adhere to the prevalent definition of model collapse in the field and align with practical considerations. We believe this perspective reflects the most realistic scenario for practitioners faced with training corpora pulluted with AI-generated content, as detailed in **Point 1**, **Point 2**, and **Point 3** above.

---

### Public Comment · ~Rylan_Schaeffer2 · 2024-12-03
**Scientific Claims Require Discussion & Citation of Intentionally-Omitted Contradictory Prior Work (Part 2)**

Dear Reviewers & Area Chairs,

I would like to again emphasize that, based on the response by the authors of Strong Model Collapse below, the authors are:

1. clearly aware of [prior published work](https://openreview.net/forum?id=5B2K4LRgmz#discussion) that directly contradicts their narrative and their scientific claims, while
2. using the methodology introduced by said prior work that they simultaneously insult and do not credit.

The authors' response demonstrates that the authors intentionally chose to not discuss these complexities, despite clearly having much to say on the topic. Rather, the authors are instead relying on a bad-faith interpretation of the ICLR reviewing guidelines to justify their intentional omission of a (reasonably) well-cited published work that demonstrates the fraction of real data can vanish without exhibiting "a critical degradation" of performance.

While the authors are well within their rights to challenge prior work, **suppressing contradictory evidence is antithetical to the scientific process and muddies the research community's understanding of the (potential) harms of model collapse.**

To make two additional comments:

1. The ICLR guidelines are meant to help authors avoid being penalized for being unaware of contemporaneous work. The authors of Strong Model Collapse are well aware of our paper since April 2nd 2024, and are even credited for their feedback in its footnotes twice. The ICLR guidelines are not meant to excuse intentionally not citing work that the authors are familiar with.

2. To be clear, I am not claiming credit for TinyStories (and we appropriately cite it). I am claiming credit for demonstrating that GPT2+TinyStories is a viable and insightful setting for studying model collapse, which is a non-trivial extension and (to the best of my knowledge) was novel. None of the 10 citations the authors cite below contain the string "model collapse".

---

### Comment · Program_Chairs · 2025-04-12

This statement clarifies the review process for #4448 ("Strong Model Collapse").

The reviewers were unanimously positive about the paper.  The AC recommended reject on the basis of a public comment stating a missing citation to a COLM 2024 paper.  Per ICLR Reviewer Guidelines, the COLM 2024 paper is within the grace period of not requiring a citation (published after July 1st, 2024).

The authors acknowledge knowing about an earlier version of the paper that was on arXiv, but chose not to cite that paper.  Per ICLR Reviewer Guidelines, authors are encouraged to cite papers that are not peer-reviewed, but it’s ultimately a judgement call (by the reviewers).

In private discussions (on OpenReview but not viewable to the public), the reviewers ultimately decided that missing this citation cannot be the sole basis for rejection.  The AC overruled the reviewers and recommended reject.  The Programs Chairs received a complaint and asked an internal committee as well as the Senior AC to review the case.  The unanimous decision is to agree with the reviewers, thus accepting the paper. Any paper that achieves an average score above a threshold is automatically considered for spotlight.

To be absolutely clear, ICLR’s decision-making is completely within this scope.  ICLR is not making any conclusions about any allegations of plagiarism or other misconduct at other venues.  Any implications by any party suggesting otherwise is false.

---

### Meta-Review · Area_Chair_myDB · 2024-12-19

**Metareview:**

This paper received unanimously positive reviews from double blind peer review. During the discussion period, a public comment was posted about the lack of citation and discussion for a related paper, first published at COLM 2024 in October 2024. While ICLR's policy exempts papers published after July 1, 2024 from citation requirements, the authors acknowledged being aware of an early conception and development of the COLM paper, but had decided not to cite it.

There was significant discussion among the authors, area chair, and reviewers for whether the authors are required to cite the paper. The program chairs consequently convened a panel of prior ICLR program chairs who weighed whether the COLM paper needed to be cited. The panel voted unanimously, given the evidence, that the authors are not required to cite the COLM paper. Since the main reason for rejection was the missing citation and otherwise received positive reviews, the paper is hence accepted.

The original meta-review for the paper is below.

-----------------

(I) The paper discusses the very relevant topic of model collapse in the presence of synthetic data generated by LLMs in the training datasets. The authors hypothesize that even a small fraction of synthetic data influences the performance of pre-trained LLMs. All reviewers agree on the quality of the manuscript in terms of being well-written and having practical findings. Furthermore, the reviewers appreciated the soundness of the theoretical results. I agree with the judgment of the reviewers.

(II) However, a comment by a community member indicated that the authors were aware of a closely related prior publication and did not cite it. Based on the author's own words (https://openreview.net/forum?id=et5l9qPUhm&noteId=k34IfU2Hvs), they deliberately decided to not cite the prior work :

"... we still don't think Gerstgrasser et al. (2024) is scientifically sound (for example, in terms of the technical contributions and most importantly, the misleading/inaccurate claims of solving/mitigating model collapse, as explained above), and therefore didn't feel the need to cite."

Based on established practices of good scientific conduct, closely related prior work should be cited and appropriately discussed. The AC concludes that no author should willingly refuse to cite a closely related prior work published at a peer-reviewed conference, even if they might disagree with the arguments, quality, or methodology of a prior work. Instead, the authors should cite the prior work and criticize it.

(III) During the discussions between the AC and the paper reviewers, an argument was raised that the ICLR policy permits not citing papers published after 1 July. In this concrete case, the committed prior work was published at COLM 2024 after 1 July.

 The applicable policy (https://iclr.cc/Conferences/2025/ReviewerGuide) states that:

"We consider papers contemporaneous if they are published within the last four months. That means, since our full paper deadline is October 1, if a paper was published (i.e., at a peer-reviewed venue) on or after July 1, 2024, authors are not required to compare their work to that paper. Authors are encouraged to cite and discuss all relevant papers, but they may be excused for not knowing about papers not published in peer-reviewed conference proceedings or journals, which include papers exclusively available on arXiv. Reviewers are encouraged to use their good judgment and, if in doubt, discuss with their area chair. “

However, the policy focuses on good faith ommissions of the prior work due to not knowing such papers:

“… Authors are encouraged to cite and discuss all relevant papers, but they may be excused for NOT KNOWING about papers not published in peer-reviewed  …, which includes papers exclusively available on arXiv.”.

However, the case of this submission is different, because the authors admitted to being aware of the work but deliberately decided to not cite it. As a result, I do not see this case as excusable under the ICLR regulation, as the latter implies a good faith omission of prior work.

(IV) As a result, I recommend rejecting the paper.

**Additional Comments On Reviewer Discussion:**

All the reviewers acknowledged and replied to the authors' rebuttal comments. There was a comment from a non-reviewer member of the community that elicited a discussion on potential misconduct in terms of deliberately committing the citation of a prior work (see the metareview).

---

> ### Comment · Area_Chair_myDB · 2025-03-23
> **Clarification**
>
> As the AC of the paper I clarify that my recommendation was to reject the paper. The decision was overturned to an Accept by the Program Chairs.

---

### Decision · Program_Chairs · 2025-01-22

Accept (Spotlight)